# Global fine-mode aerosol radiative effect, as constrained by comprehensive observations

Chul E. Chung[1], Jung-Eun Chu[2], Yunha Lee[3], Twan van Noije[4], Hwayoung Jeoung[5], Kyung-Ja Ha[2], Marguerite Marks[6]

[1]Desert Research Institute, Reno, 89512, USA
[2]Pusan National University, Busan, 46241, Korea
[3]Washington State University, Pullman, 99164, USA
[4]Royal Netherlands Meteorological Institute, AE De Bilt, 3730, Netherlands
[5]National Meteorological Satellite Center, 27803, Korea
[6]Carnegie Mellon University, Pittsburgh, 15213, USA

*Correspondence to*: C. E. Chung (Eddy.Chung@dri.edu)

**Abstract.** Aerosols directly affect the radiative balance of the Earth through absorption and scattering of solar radiation. Although the contributions of absorption (heating) and scattering (cooling) of sunlight have proved difficult to quantify, the consensus is that anthropogenic aerosols cool the climate, partially offsetting the warming by rising greenhouse gas concentrations. Recent estimates of global direct anthropogenic aerosol radiative forcing (i.e., global radiative forcing due to aerosol-radiation interactions) are $-0.35\pm0.5$ Wm$^{-2}$, and these estimates depend heavily on aerosol simulation. Here, we integrate a comprehensive suite of satellite and ground-based observations to constrain total AOD, its fine-mode fraction, the vertical distribution of aerosols and clouds, and the co-location of clouds and overlying aerosols. We find that direct fine-mode aerosol radiative effect is $-0.46$ Wm$^{-2}$ ($-0.54 \sim -0.39$ Wm$^{-2}$). Fine-mode aerosols include sea salt and dust aerosols, and we find that these natural aerosols pose a very large cooling ($-0.44 \sim -0.26$ Wm$^{-2}$) when constrained by observations. When the contribution of these natural aerosols is subtracted from the fine-mode radiative effect, the net becomes $-0.11$ ($-0.28 \sim +0.05$) Wm$^{-2}$. This net arises from total (natural + anthropogenic) carbonaceous, sulfate and nitrate aerosols, which suggests that global direct anthropogenic aerosol radiative forcing be less negative than $-0.35$ Wm$^{-2}$.

## 1. Introduction

Atmospheric aerosols absorb and scatter solar radiation and act as cloud condensation nuclei, thus affecting cloud albedo and lifetime. The climatic effect of anthropogenic aerosols is usually quantified in terms of radiative forcing, defined as the net radiative flux perturbation at the top of the atmosphere (TOA) owing to aerosol changes since pre-industrial time to the present. The magnitude of aerosol radiative forcing is recognized as the most uncertain component of estimated total radiative forcing (Myhre et al., 2013a). The magnitude of the global average of aerosol direct radiative forcing (which is referred to as radiative forcing due to aerosol-radiation interactions in the 5th IPCC report) has been estimated to range from $-0.85$ to $+0.15$ Wm$^{-2}$ (Myhre et al., 2013a).

Direct aerosol forcing has been commonly estimated by a model-based approach of simulating global aerosol amount, distribution, and characteristics, and processing the predicted global aerosol distribution by a radiation model. Global aerosol simulations are subject to large uncertainties in emissions, transport, gas-to-aerosol

conversion, aerosol aging, aerosol mixing state, and wet and dry deposition (Bond et al., 2004; Ma et al., 2012). The
large spread among direct aerosol forcing estimates (Myhre et al., 2013a) is attributable largely to these simulation
uncertainties. Plus, processing the calculated aerosol distribution by a radiation model requires the specification of
parameters such as the single scattering albedo (SSA) of organic aerosol which has been treated as 0.96~1.0 at 550
nm in the modeling community (Myhre et al. 2013b) but might actually be much lower (e.g., 0.85 estimated by
Magi 2009; 2011). Attempts have been made to bypass some of these uncertainties and constrain calculated aerosol
optical properties by observations (Chung et al., 2005; Bellouin et al., 2008; Myhre, 2009; Su et al., 2013) but these
semi-empirical studies are not sufficient to validate the model based estimates given heavy model dependence. In
particular, the anthropogenic fraction of aerosol amount was obtained entirely from aerosol simulations (Chung et al.,
2005; Myhre, 2009; Su et al., 2013) or by utilizing the fine-mode fraction (FMF) of satellite-derived AOD (Aerosol
Optical Depth) over ocean (Bellouin et al., 2008; Chung et al., 2005). Over the land, where most anthropogenic
aerosols are located, no study constrained the anthropogenic fraction by observations yet.
Aerosols have different sizes, and typically follow a bimodal structure in terms of fine mode and coarse
mode (Kim et al., 2007; Viskari et al., 2012). Fine-mode aerosols usually have submicron sizes in diameter and
these small particles are mostly anthropogenic. In this study, we provide observational estimates of direct fine-mode
aerosol radiative effect (i.e., anthropogenic + natural forcing due to all the fine-mode aerosols). In particular, we
constrain total AOD, SSA, and the asymmetry parameter by observations as in previous semi-empirical studies
(Chung et al., 2005; Myhre, 2009; Su et al., 2013). In addition, we use observations to constrain the aerosol vertical
profile, and the FMF of AOD over land as well as ocean. There is some use of simulated aerosol to fill up
observation gaps in our study but the use is highly limited, and we address the uncertainty due to the use of
simulation. When our observational estimates are compared to the simulated fine-mode aerosol radiative effects, one
can obtain additional insights into biases and uncertainties in the aerosol forcing estimates from aerosol simulations.
Atmospheric aerosols consist of carbonaceous, sulfate, nitrate, sea salt and dust aerosols. The first three
types of aerosols are fine-mode particles which are mostly anthropogenic while a sizable portion of sea salt and dust
aerosols are also in the fine mode. Thus, offering observational estimates of fine-mode aerosol radiative effect is an
important advance but is not sufficient in understanding the biases in the aerosol forcing estimates from aerosol
simulations. In the present study, we will use observations to constrain the fine-mode sea salt and dust AODs as
well, and offer estimates of aerosol radiative effect due to fine-mode sea salt and dust aerosols.
**2. Data**
In section 4 and Table 1, aerosol direct radiative effects (DRE) will be computed for three cases: (i) for
total aerosols, (ii) for the fine mode (including natural fine-mode particles), and (iii) for fine mode sea-salt and dust.
The total and fine-mode AOD are based on observations, as explained in Section 2.1. The other aerosol optical
properties needed for the DRE calculations are derived as follows:
• The asymmetry parameter (ASY), SSA and the Co-albedo Ångström Exponent (CAl_AE) for the total aerosols are
derived by nudging GOCART simulated values towards AERONET data (Section 2.2). The spectral dependence of
ASY is addressed as in Chung et al. (2005).
• The fine-mode aerosol DRE is computed as the difference between the total and coarse mode DREs. The coarse-
mode ASY, SSA and CAl_AE are derived from GOCART simulations, as explained in Section 2.3.
• For computing the DRE due to fine-mode sea-salt and dust, ASY, SSA and CAl AE are derived from GOCART
simulations (Section 2.3).

The datasets used to derive this information are explained in the following. All the datasets used in this

study are monthly means.

## 2.1 Global observational data

AOD is a common measure of aerosol amount. AERONET (Aerosol Robotic Network; Holben et al., 1998)

AOD is known to be the most accurate global-scale product. However, AERONET sites are non-uniformly
distributed over the globe while less-reliable satellite (MODIS and MISR) AODs have nearly full global coverage.
We follow the approach of Chung et al. (2005) and Lee and Chung (2013) in nudging or adjusting the satellite AOD
towards AERONET AOD to construct globally-reliable AOD from 2001 to 2010. See Chung et al. (2005) and Lee
and Chung (2013) for the visual effects of the nudging. Fig. 1A shows this adjusted AOD. Also, AOD Ångström
exponent from 2001 to 2010 is derived by adjusting the satellite data towards AERONET data as in Lee and Chung

(2013).

Fine-mode AOD (fAOD) at 500 nm from 2001 to 2010 are obtained by the approach in Lee and Chung

(2013), except that instead of directly using the monthly AERONET FMF data we used the monthly AERONET
fAOD (from the direct sun measurements and the Spectral Deconvolution Algorithm as in Lee and Chung) and total
AOD to derive the FMF. Like in Lee and Chung (2013), we convert AOD Ångström exponent data into FMF data,
and nudge this FMF data towards AERONET FMF data to derive reliable FMF and thus fine-mode AOD over the
globe. Note that the definition of fine mode in the present study thus follows that by the AERONET Spectral
Deconvolution Algorithm as in O'Neill et al. (2003) and Lee and Chung (2013). Coarse-mode AOD at 500 nm is
obtained by subtracting fine-mode AOD from total AOD at 500 nm.

We computed the 2001-2010 average for each calendar month at the T42 resolution. In these datasets, the

observational data gaps are filled by the GOCART simulation (Chin et al., 2002) as in Lee and Chung (2013).
These data gaps are predominantly confined to the polar regions, and are even fewer in polar summer.

We obtain fAOD at 550 nm by subtracting coarse-mode AOD at 500 nm from AOD at 550 nm, assuming

that coarse-mode AOD does not change from 500 nm to 550 nm. That is, $fAOD_{550} = fAOD_{500} + (AOD_{550} -$
$AOD_{500})$. Total AOD at any wavelength is obtained by combining AOD at 550 nm and AOD Ångström Exponent.

## 2.2 Global semi-observational data

To compute the direct aerosol radiative effect, aerosol optical characteristics, such as SSA (Single

Scattering Albedo), must be specified. We construct a global distribution of SSA by nudging global model-
simulated (Chin et al., 2002) SSA towards AERONET SSA. We apply a similar procedure to ASY.
550 nm SSA, 550 nm ASY, CAl_AE (Co-albedo Ångström Exponent; Co-albedo = 1-SSA) for total
(natural + anthropogenic) aerosols are obtained by nudging the GOCART simulation (Chin et al., 2002) towards
AERONET data.  Specifically, for ASY and CAl_AE the following nudging equation is used:
$\text{N\_ASY}_j = G_{ASY_j} + \dfrac{\sum_i \frac{AERONET\_ASY_i - G\_ASY_i}{d_{j,i}^4}}{\sum_i \frac{1}{d_{j,i}^4}}$        (1)
where N_ASY$_j$ is the adjusted new value of ASY at grid j; AERONET_ASY$_i$ is an AERONET ASY at station i; $d_{j,i}$
is the distance between j and i; and G_ASY$_i$ is the GOCART ASY at the grid box containing AERONET$_i$. Here the
AERONET data and the GOCART simulation are on the T42 grids.
For SSA, the following equation is used:
$(1 - \text{N\_SSA}_j) = (1 - G\_SSA_j) \times \dfrac{\sum_i \frac{1 - AERONET\_SSA_i}{d_{j,i}^4}}{\sum_i \frac{1 - G\_SSA_i}{d_{j,i}^4}}$     (2).
Another way to interpret the above equations (Eq. 1, 2) is that the GOCART simulation is an interpolation tool.  The
equation for SSA differs from that for ASY or CAL_AE, because SSA cannot be negative, and its value goes down
from 1.0.  The above equations are applied for each grid and each calendar month.  The final values are the
simulation nudged towards AERONET values.  Please note that these above equations (Eq. 1 & 2) were also used in
Chung et al. (2005) but a clear explanation was not given in that study.
Before combining the GOCART simulation and AERONET data, the 2001-2010 average was calculated
from the monthly Level 2.0 AERONET data for each calendar month.  The average of the AERONET SSA or ASY
was AOD-weighted. Then, SSA and ASY at 550 nm were obtained from the neighboring wavelength values through
linear interpolation.  AERONET CAl_AE was obtained by the 2001-2010 SSA averages at 440, 675 and 870 nm for
each calendar month.  Please note that AERONET only gives level 2 quality SSA when AOD at 440 nm >0.4, and
therefore many regions of the earth do not have AERONET SSA data.
The products from combining the GOCART simulation and AERONET data are semi-observational and
we address the model dependence as follows.
SSA, apart from AOD, is the most influential parameter in aerosol direct forcing (Chung, 2012).  We first
generated three different sets of simulated SSA:
SSA1 = (0.19×BC_AOD+0.85×OA_AOD+1.0×sulfate_AOD+1.0×sea-salt_AOD+0.96×dust_AOD)/total_AOD;

SSA2 = (0.14×BC_AOD+0.8×OA_AOD+1.0×sulfate_AOD+1.0×sea-salt_AOD+0.96×dust_AOD)/total_AOD; and

SSA3 = (0.19×BC_AOD+0.98×OA_AOD+1.0×sulfate_AOD+1.0×sea-salt_AOD+0.96×dust_AOD)/total_AOD.

BC_AOD above refers to the GOCART BC AOD at 550 nm.  We chose parameters (e.g., 0.19 for BC SSA) in the
above three SSA equations from various observational studies (e.g., Magi, 2009; Magi, 2011).  Additionally, in
SSA2 (more absorbing case), we doubled the magnitude of BC AOD, given a notion (e.g., Chung et al., 2012) that
simulated BC is significantly underestimated.  We use the above three sets of simulated SSA in order to produce an
initial estimate of the uncertainty in simulated SSA. Then, we nudged the 3 sets of simulated SSA towards the same
AERONET SSA, which gave 3 sets of semi-observational SSA. Finally, we computed the average, maximum and
minimum SSAs from the three sets of SSA over each grid and each calendar month, and then re-generated three sets
of SSA (average: baseline; maximum; and minimum; see Fig. 2). This re-generation increases the global-average
SSA difference between the least absorbing and most absorbing cases. We do this re-generation in an attempt to
fully bracket the simulated SSA uncertainty. The last procedure (i.e., re-generation) assures that the final three sets
of SSA depend insignificantly on the initial estimate of the simulated SSA uncertainty.
Simulated ASY at 550 nm and CAl_AE are computed as follows.
ASY = (0.6×CA_SAOD+0.7×sulfate_SAOD+0.75×sea-salt_SAOD+0.75×dust_SAOD)/total_SAOD.
CA_SAOD here refers to Carbonaceous Aerosol (i.e., BC+OA) SAOD (Scattering AOD) at 550 nm from GOCART.
CAl_AE = (-0.53×CA_AOD+2.215×dust_AOD)/total_AOD,
where CA_AOD refers to CA AOD. The chosen parameters (e.g., 2.215) in the ASY and CAl_AE equations are
from preliminary AERONET data analysis. These simulated ASY and CAl_AE were nudged towards AERONET
data as explained earlier. We do not address the model dependence on ASY or CAl_AE, since its impact on aerosol
forcing is tiny compared to the impact of SSA uncertainty. To be sure, we re-generated the ASY using doubled BC
AOD while holding other components (such as SSA) fixed, and found that the global direct aerosol effect changes
by less than 0.002 Wm$^{-2}$.
The GOCART simulations were prepared as follows. We used sea salt AOD from Chin et al. (Chin et al.,
2002), and BC (black carbon), OA (Organic Aerosol), dust and sulfate AODs from the Giovanni website
(http://gdata1.sci.gsfc.nasa.gov/daac-bin/G3/gui.cgi?instance_id=neespi), which contains GOCART model output
from 2000 to 2007. These AODs are monthly means at 550 nm. Then, the climatological seasonal cycle for the
available data period was computed. We used these simulated AOD values to compute the simulated SSA, ASY
and CAL_AE.
**2.3 Global simulations**
For coarse-mode aerosols, we assumed ASY to be 0.75 and AOD Ångström exponent to be 0.0. For 550
nm SSA and CAL_AE, we rely entirely on the GOCART simulations as follows: SSA = (1.0×sea-
salt_AOD+0.96×dust_AOD)/lAOD, where dust_AOD refers to GOCART dust AOD and lAOD refers to GOCART
dust and sea salt AODs combined. CAL_AE=2.215×dust_AOD/lAOD. Although we rely entirely on simulated
SSA for coarse-mode aerosols, we find very small the coarse-mode aerosol radiative effect uncertainty resulting
from simulated SSA. For instance, when we change the dust AOD by 35%, the difference in coarse-mode aerosol
radiative effect is only 0.01 Wm$^{-2}$.
For fine-mode sea salt and dust aerosols, we assumed ASY to be 0.6 and AOD Ångström exponent to be
1.85. For 550 nm SSA and CAL_AE, we rely entirely on simulated fAODs as follows:
SSA = (1.0×sea-salt_fAOD+0.96×dust_fAOD)/fAOD,
where dust_fAOD refers to dust fAOD and fAOD refers to dust and sea salt fAODs combined.
CAL_AE=2.215×dust_fAOD/lAOD.

These simulated aerosol optical properties were used in the MACR model runs, leading to the results in

Table 1.
**2.4 Vertical profile**

Aerosol vertical profiles are obtained from the space-borne CALIOP lidar (Liu et al., 2009). To construct

the profile, we used the daytime CALIPSO lidar level 2.0 data (Liu et al., 2009) from June 2006 to Oct. 2011. We
processed the level 2.0 data, and obtained clear-sky aerosol extinction coefficient at 532 nm at the T42 spatial
resolution and 500 m MACR model vertical resolution. We filled the data gaps using available neighboring data
through linear interpolation. We then computed the climatological seasonal cycle for the entire available data period.
Over some grids and calendar months, the aerosol extinction coefficient has extremely low magnitudes, in which
case, the PBL profile as in Chung et al. (Chung et al., 2012) is applied. The threshold for applying the PBL profile
is a vertically-summed aerosol extinction coefficient of 0.03. Note that a vertically-summed aerosol extinction
coefficient of 0.03 is associated with a very small amount of aerosol and the effect of these aerosols on global
aerosol direct effect is very small. Also note that the aerosol vertical profile from CALIPSO is scaled to match the
AOD observations obtained by integrating AERONET, MODIS and MISR data (as shown in Fig. 1A) since the
latter observations describe clear-sky AOD too and give better accuracy. The clear-sky aerosol profile from
CALIPSO is assumed to be applied to an entire T42 grid in the MACR model.

To adjust the magnitude of AOD over cloud by CALIPSO data, we use the daytime CALIPSO lidar level

3.0 data (Winker et al., 2013), which are globally-gridded ($5° \times 2°$) monthly mean data spanning from June 2006 to
Jan. 2012. Specifically, we use the CALIPSO level 3.0 derived ratio of clear-sky AOD to above-cloud AOD to
modify the aerosol amount over cloud over each grid cell in the MACR model. The level 3.0 data have gaps. Again,
the data gaps were filled using a linear interpolation, then the data was converted into the T42 grids, and the
climatological seasonal cycle was obtained before the use in MACR model.

For coarse-mode aerosols, we apply the same profiles given a lack of observations. Because coarse-mode

aerosols are not very absorbing, the effect of the vertical profile is very small (see Choi and Chung, 2014).
**3. Radiation model**

We use the Monte-Carlo Aerosol Cloud Radiation (MACR) model as in Choi and Chung (2014), except

that we improved the low cloud height in the model using the CALIPSO level 2.0 data. As in Choi and Chung
(2014), the height of low cloud bottom is set to 750 m above the ground. The low cloud top height is set to 1250 m,
when the maximum low cloud height over a $5° \times 2°$ grid (and during a whole month) from CALIPSO data is 750m ~
1750m. When the CALIPSO maximum low cloud height exceeds 1750 m, the low cloud top height in the model is
set to 1750 m above the ground.

This model was built upon the so-called Monte Carlo Independent Column Approximation (McICA)

approach (Pincus et al., 2003); uses a set of satellite observations to describe multi-layer cloud, surface albedo, and
stratospheric column ozone; and uses ERA-Interim Reanalyses (Dee et al., 2011) to describe the precipitable water.
An earlier version has undergone comprehensive validation of the simulated fluxes at the TOA and at the surface
over 100 land and island stations (agreement with observations is within a few $Wm^{-2}$) (Kim and Ramanathan, 2008).
Only short-wave radiation is considered here.
**4. Aerosol direct radiative effect**
We first address the direct aerosol radiative effect (forcing due to natural and anthropogenic aerosols). We
incorporated the integrated global aerosol data (as explained in section 2) into the MACR model. Fig. 1B shows
the direct aerosol radiative effect as estimated by the MACR model. The direct aerosol radiative effect in Fig. 1B
also incorporates that aerosol amount over cloud might differ from that at the same height in clear skies in the same
region. The CALIOP lidar is able to retrieve aerosol amount over cloud as well as in clear skies, and so we used this
lidar data to constrain the aerosol amount over cloud (as explained in section 2.4) in computing the direct aerosol
radiative effect. This procedure could be important since radiation modeling studies showed that the sensitivity of
aerosol forcing to the aerosol vertical profile arises mainly as a consequence of the location of absorbing particles
relative to cloud (Choi and Chung, 2014). On the other hand, cloud is brighter than most surfaces during daytime,
and this could create a low bias in aerosol amount over cloud, as retrieved by the CALIOP lidar (Chepfer et al., 2013;
Hunt et al., 2009; Kacenelenbogen et al., 2014; Vaughan and coauthors, 2009). To be sure, we re-computed the
aerosol radiative effect assuming equal amounts between clear skies and over cloud, and found that the radiative
effect only increases by 0.03 $Wm^{-2}$ in global average.
Next, we estimate fine-mode aerosol radiative effect. Since the FMF of aerosols over land is difficult to
accurately retrieve from satellites, past semi-empirical estimates (Bellouin et al., 2008; Myhre, 2009) only used the
FMF of AOD from satellite observations over the ocean. In contrast, AERONET data provide relatively reliable
FMF over both land and ocean (with the AERONET data being predominantly over land). Following the approach
of Lee and Chung (2013) satellite data are nudged toward AERONET data to construct global FMF and thus fine-
mode AOD (see section 2.1 for details). Fig. 3A shows this fine-mode AOD, which, as expected, is largest over
industrial and biomass burning areas.
Fig. 3B shows the estimated fine-mode direct radiative effect as the difference between the coarse-mode
and total (coarse + fine modes) aerosol radiative effect. Fine-mode radiative effect is negative almost everywhere,
except over the eastern equatorial Atlantic, the Sahara, and the Arabian Desert. These areas of positive forcing
result from highly absorbing particles above highly reflective surfaces or low cloud. The global average of the fine-
mode direct radiative effect is estimated as −0.46 $Wm^{-2}$. In this computation, aerosol simulation using GOCART
was used to provide interpolation for aerosol optical characteristics, such as SSA. To quantify uncertainty in the
model dependence, two sets of additional simulations were conducted, representing lower and upper limits of
absorption efficiency (see section 2.2 and Fig. 2). Fine-mode radiative efffect is estimated to range between −0.54
$Wm^{-2}$ and −0.39 $Wm^{-2}$, corresponding to these two limits (Table 1). Aerosol simulations yielding fine-mode
radiative effect outside of the −0.54 ~ −0.39 Wm$^{-2}$ range can be considered as inconsistent with observational
constraints.
**5. Fine-mode fraction (FMF) of sea salt and dust AODs**
The fine-mode direct radiative effect estimate, as shown in Fig. 3B, includes the contribution from natural fine-
mode sea salt and dust aerosols. To subtract this contribution from the fine-mode direct radiative effect estimate, we
address the FMF of sea salt and dust AODs here. Instead of using simulated fine-mode sea salt and dust AOD (and
thus being 100% subject to model uncertainties), we use observed coarse-mode AOD $\times \frac{SD\_FMF}{1-SD\_FMF}$, where SD_FMF
refers to the simulated FMF of sea salt + dust AOD. An underlying assumption therein is that coarse-mode AOD
results only from sea salt and dust aerosols. We obtain the observed coarse-mode AOD by subtracting fine-mode
AOD from total AOD where the fine-mode and total AODs were obtained by integrating AERONET, MODIS and
MISR data (see section 2.1). On rare occasions, $\frac{SD\_FMF}{1-SD\_FMF}$ becomes unrealistically large. To prevent this, we limit
fine-mode sea salt and dust AOD to be < 99% of total fine-mode AOD.
For simulated FMF, we used AOD (at 550 nm) simulations from GOCART, the Spectral Radiation-Transport
Model for Aerosol Species (SPRINTARS), the Tracer Model 5 (TM5) and ModelE2-TOMAS (briefly ModelE2
here). The SPRINTARTS output is from the AeroCom (Aerosol Comparisons between Observations and Models)
Phase II (Schulz et al., 2009) hindcast experiments and the TM5 outputs are from the AeroCom Phase III. The
ModelE2-TOMAS simulation was performed using the TwO-Moment Aerosol Sectional (TOMAS) microphysics
module incorporated into the state-of-the-art general circulation model GISS ModelE2 (Lee et al., 2015). TOMAS
module represents aerosol size distribution in many size categories or "bins" covering 10nm to 10µm. We used a
Fast-TOMAS module (Lee and Adams, 2012) with a 15 bin version here, since Fast TOMAS reduces the
computational burden by 2-3 times while well preserving the capability of computing fine-mode fraction compared
to the original TOMAS model with 30 bins. The fine-mode fraction of dust and sea-salt aerosols from ModelE2-
TOMAS was calculated by converting the mass output to AODs, and then applying the Spectral Deconvolution
Algorithm (SDA) used in AERONET retrievals (O'Neill et al., 2003) to the AODs in order to create FMF consistent
with AERONET FMF. A Mie-scattering code was used to compute size-resolved AOD at 380, 440, 500, 675 and
870 nm. Refractive indices for dust and sea-salt are taken from Optical Properties of Aerosol and Clouds (OPAC)
dataset (Hess et al., 1998). For other models, we calculated FMF using AODs from fine-mode aerosols and coarse-
mode aerosols.
The ModelE2-TOMAS simulation was nudged with wind from MERRA (Mordern Era Retrospective-analysis
for Research and Applications) reanalysis from 2003 to 2005 after 3 years of spin-up. The simulation period for
ModelE2-TOMAS is 2003-2005, and that for TM5, SPRINTARS and GOCART are 2001-2010, 2001-2008, and
2000-2007, respectively. Climatological AODs for each of 4 models were obtained by computing the average over
the aforementioned simulation period for each calendar month.
Fig. 4 is displayed to compare various simulated FMFs with the observed FMF. First, we assess which
simulation performs the best in simulating dust FMF by looking at the simulated FMFs (including FMF of non-dust
particles) over dust dominated places where we use AERONET observations to validate the simulated FMFs. Dust-
dominated AERONET sites in Fig. 4A were selected with the following criteria: 550 nm FMF < 0.3, AAE
(Absorption Ångström Exponent) > 2.0 and 550 nm AAOD (Absorption AOD) > 0.03. We again followed the
approach by Lee and Chung (2013) in computing AERONET FMF, AAE and AAOD. Please note that in Fig. 4 we
used climatological means for each calendar month; again for FMF we used mean AODs to compute the FMF
instead of averaging FMFs. Fig. 4B suggests that models tend to over-estimate dust FMF, at least over dust-
dominated places, as previously pointed out by Kok (2011).
Regarding sea salt FMF, we look at the simulated sea salt FMFs and observed total FMF over relatively pristine
oceans (Fig. 4C). Organic and sulfate aerosols can be over remote oceans (Shank et al., 2012) in addition to fine-
mode sea salt. Fig. 4C shows large disagreements between sea salt FMF simulations, where one of the models (i.e.,
GOCART) clearly overestimates sea salt FMF given that the simulated sea salt FMF is near the total FMF from
observations. In view of this, we scale down the simulated fine-mode dust FMF and mix sea salt FMF simulations
to calculate FMF of sea salt.
We scale down the simulated dust FMF and mix sea salt FMF simulations by having multiple estimates (best
estimate and sensitivity runs) to address the uncertainty in simulated FMF. The FMF of sea salt + dust AOD for our
best estimate (i.e., baseline) is prepared using ModelE2 as follows. We scale up the coarse-mode dust AOD by 1.16
times and scale down the fine-mode dust AOD by 0.56 times so that ModelE2 would match AERONET FMF and
AOD over dust-dominated sites. We scale down sea salt AOD (both fine and coarse modes) by 0.6 times so that the
total AOD from ModelE2 matches AERONET data over sea salt dominated sites. We use ModelE2 for the best
estimate since this model has an advanced size distribution description and uses the SDA to divide the AOD into
fine-mode and coarse-mode components. For sensitivity run 1, we replace the ModelE2 dust AOD by the GOCART
dust AOD where the coarse-mode dust AOD is scaled up by 1.3 times and the fine-mode dust AOD is scaled down
by 0.74 times. For sensitivity run 2, we use the baseline set-up except that for sea salt AOD we equally mix the
outputs from GOCART, TM5 and ModelE2.
Scaling the simulated dust FMF to match AERONET FMF over dust-dominated sites may still have an
overestimation or underestimation of dust FMF outside of dust dominated regions. Plus, dust-dominated regions
have non-dust particles, and thus the scaled dust FMF might still underestimate or overestimate dust FMF even over
dust dominated regions. This is why we conduct sensitivity runs even after the scaling of the simulated dust FMF.

## 6. Implications for global direct aerosol radiative forcing

We estimate the direct radiative effect due to fine-mode sea salt and dust aerosols at −0.35 (−0.44 ~ −0.26)
Wm$^{-2}$ (Table 1). The spatial pattern is shown in Fig. 5. As mentioned in section 5, our estimate of fine-mode sea
salt and dust aerosols might be too large or too small over some areas. Possible over-estimation or under-estimation
is likely reduced in global average, and so we focus on global averages as shown in Table 1. The global direct
radiative effect of −0.35 $Wm^{-2}$ is quite large. In those studies where fine-mode sea salt and dust aerosols were
assumed to be negligible, the aerosol direct forcing estimates would have been that much more negative than in
reality.
When we remove the contribution of fine-mode sea salt and dust aerosols from the fine-mode radiative
effect, we end up with aerosol radiative effect due to total (i.e., anthropogenic + natural) carbonaceous, sulfate and
nitrate aerosols. As Fig. 6A shows, this radiative effect is large and positive over Africa and the downstream areas
where biomass burning is the major source. The forcing is also conspicuously positive over the Sahara (Fig. 6A),
partly because biomass burning aerosols in the Sahel are advected northwards in boreal winter (Haywood et al.,
2008) and bright desert surfaces turn the forcing positive. Fig. 6B shows that these advected aerosols have a
relatively small forcing in the atmosphere due to smaller aerosol amounts. Outside of Africa and the downstream
areas, the forcing is a mixture of positive and negative values, and negative values slightly outweigh positive values.
The global average (including Africa) of the TOA forcing (as shown in Fig. 6A) is −0.11 $Wm^{-2}$ with an uncertainty
range of −0.28 ~ +0.05 $Wm^{-2}$ which results from −0.54+0.26 ~ −0.39+0.44 $Wm^{-2}$.
The consensus of global aerosol direct radiative forcing as shown in the 5[th] IPCC report is −0.35 $Wm^{-2}$
(Myhre et al., 2013a), and this includes a dust forcing of −0.10 $Wm^{-2}$. Thus, the IPCC estimate is that
anthropogenic carbonaceous, sulfate and nitrate aerosols pose a radiative forcing of −0.25 $Wm^{-2}$, while our
observational estimate of total (anthropogenic + natural) carbonaceous, sulfate and nitrate aerosol forcing is −0.11
$Wm^{-2}$. The anthropogenic fraction (or pre-industrial fraction) of carbonaceous, sulfate and nitrate aerosols is
uncertain. Black carbon, the only warming aerosol species in carbonaceous aerosol (black carbon + organic aerosol),
sulfate and nitrate aerosol is known to be more anthropogenic than organic aerosols are (Bond et al., 2011). If the
anthropogenic fraction of black carbon is similar to that of nitrate and sulfate aerosol, the aerosol direct radiative
forcing becomes > −0.11 $Wm^{-2}$ in our observational estimation, which means that aerosol direct forcing is less
negative than the consensus as expressed in the 5[th] IPCC report.
Our observational approach makes the results subject to observation errors. AERONET SSA, in particular,
is subject to potentially significant uncertainties due to various assumptions used in the retrieval algorithms. Thus,
the uncertainty in our estimates of fine-mode forcing, e.g., might be larger than −0.54 ~ −0.39 $Wm^{-2}$. However,
studies (Eck et al., 2010; Leahy et al., 2007) showed that AERONET SSA is higher or lower than in-situ
measurements depending on location, season, in-situ measurement device, etc. Furthermore, in-situ measurements
are also subject to uncertainties, and so the difference between the AERONET SSA and in-situ measured SSA is not
necessarily due only to the AERONET data error. Overall, we believe that AERONET observations likely have
smaller biases and provide more credible results than aerosol simulations. At least, our observational approach
offers an independent estimate than pure aerosol simulations.
**Acknowledgment**
The authors are thankful to P. Adams of Carnegie Mellon University and N.T. O'Neill at Canadian Network for the
Detection of Atmospheric Change for valuable inputs. This study was funded by the National Science Foundation
(AGS-1455759).

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

**Tables**

| Direct aerosol radiative effect | Run | TOA (in Wm$^{-2}$) | Atmosphere | Surface |
|---|---|---|---|---|
| (Total) Direct aerosol radiative effect | | -2.28 | +4.77 | -7.05 |
| Fine-mode radiative effect | baseline | -0.46 | +3.88 | -4.33 |
| Fine-mode radiative effect | sensitivity run 1: least absorbing case | -0.54 | +3.63 | -4.17 |
| Fine-mode radiative effect | sensitivity run 2: Most absorbing case | -0.39 | +4.08 | -4.47 |
| Fine-mode sea salt and dust radiative effect | baseline: ModelE2 with reduced dust FMF | -0.35 | +0.23 | -0.58 |
| Fine-mode sea salt and dust radiative effect | sensitivity run 1: reduced GOCART dust FMF + ModelE2 sea salt FMF | -0.26 | +0.16 | -0.42 |
| Fine-mode sea salt and dust radiative effect | sensitivity run 2: Reduced ModelE2 dust FMF + ModelE2/GOCART/TM5 mix sea salt FMF | -0.44 | +0.26 | -0.70 |
| Fine-mode radiative effect without dust and sea salt | baseline | -0.11 | +3.64 | -3.75 |


**Table 1.** Global 2001-2010 average of aerosol radiative effect calculated with the MACR model. In this table,
natural aerosol radiative effects are included. All the aerosol radiative effect estimates made by the MACR model in
this study include 3D cloud effects.


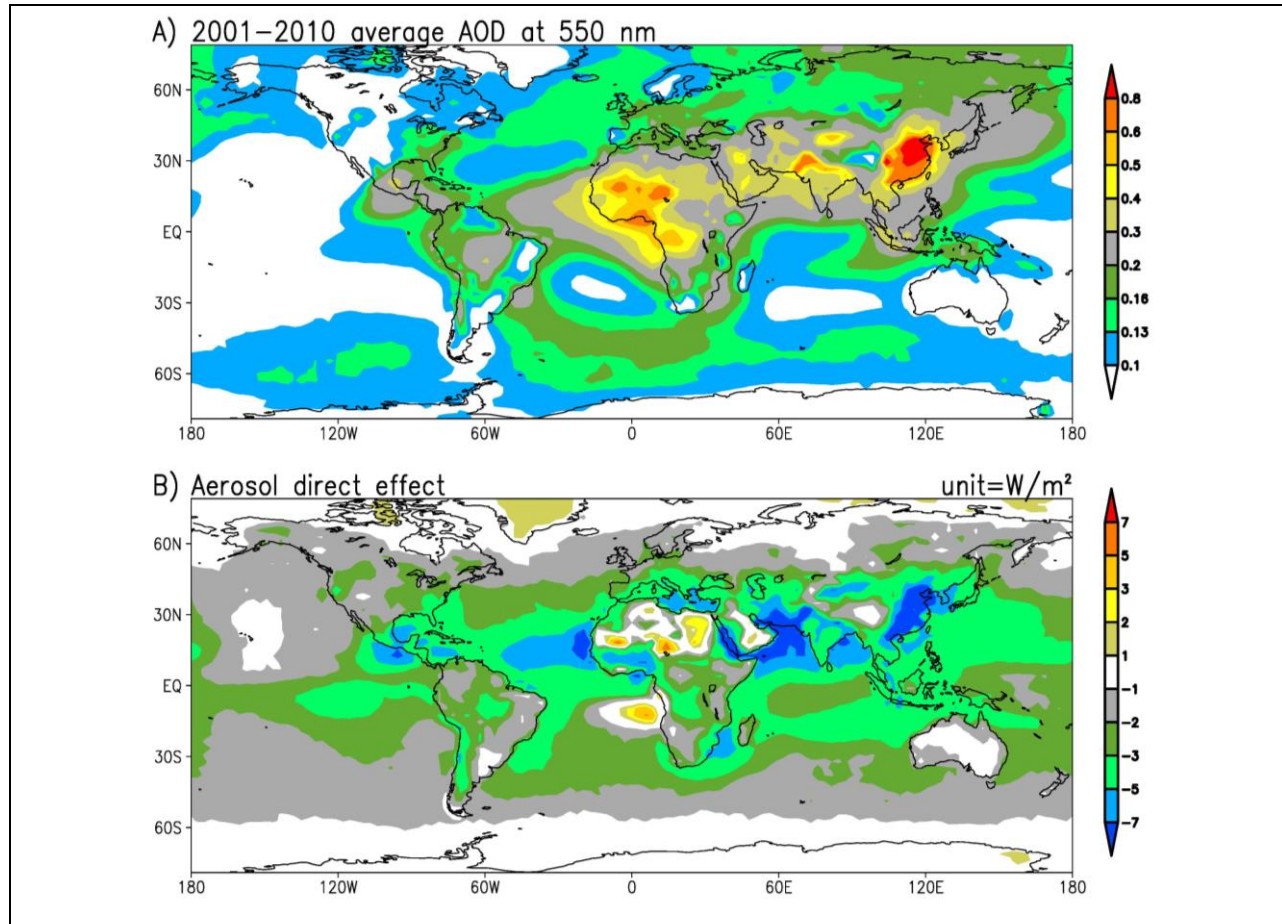

**Figure 1.** A) 2001-2010 mean 550 nm AOD obtained by integrating MODIS, MISR and AERONET AOD.  B) 2001-2010 mean direct aerosol radiative effect at TOA, as estimated by a radiation model that includes observationally-derived surface albedo.  The aerosol radiative effect estimate here includes natural aerosols.


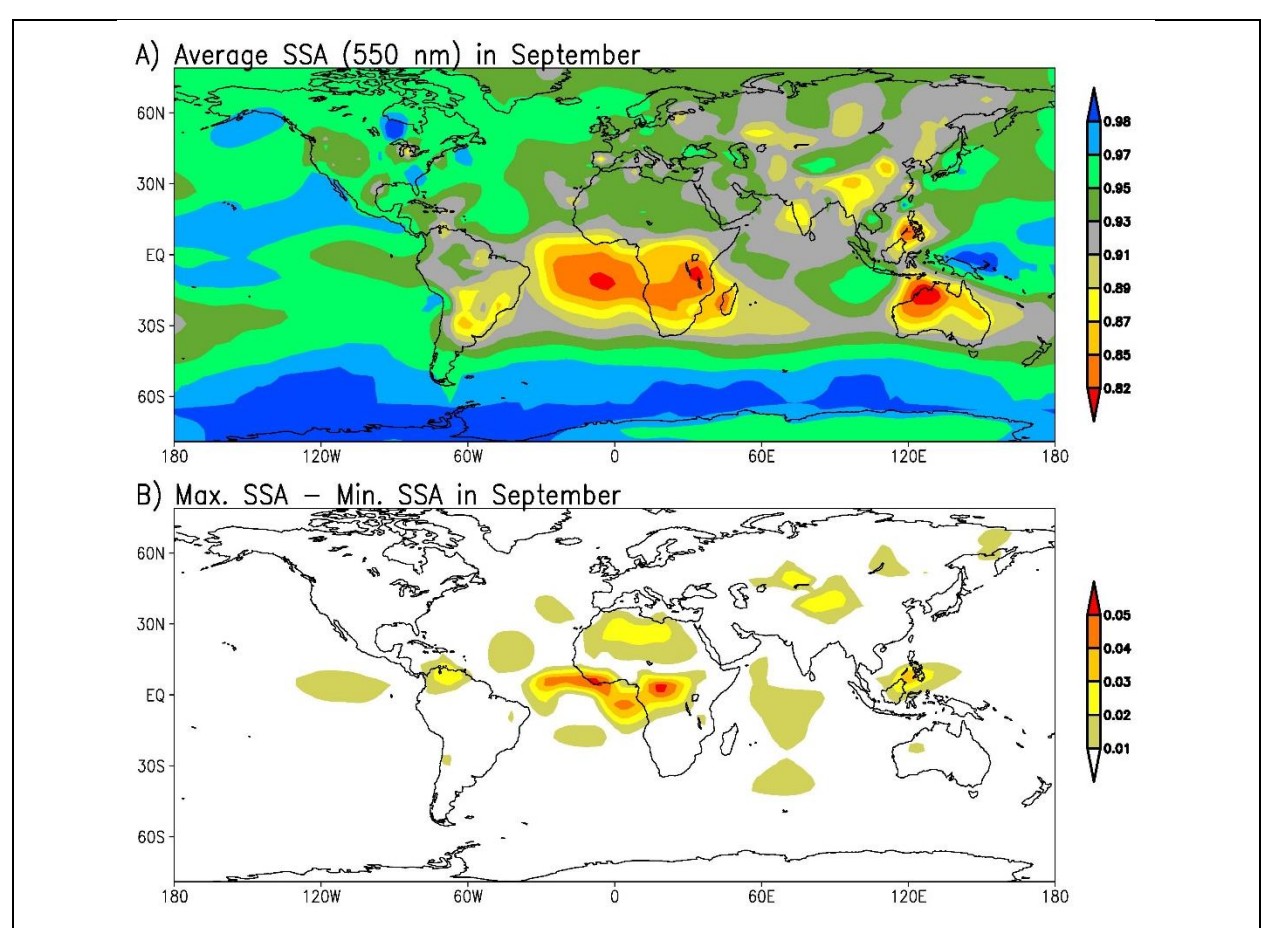

**Figure 2.** Integration of simulated SSA and AERONET SSA. A) Average SSA. B) Maximum SSA – minimum SSA.

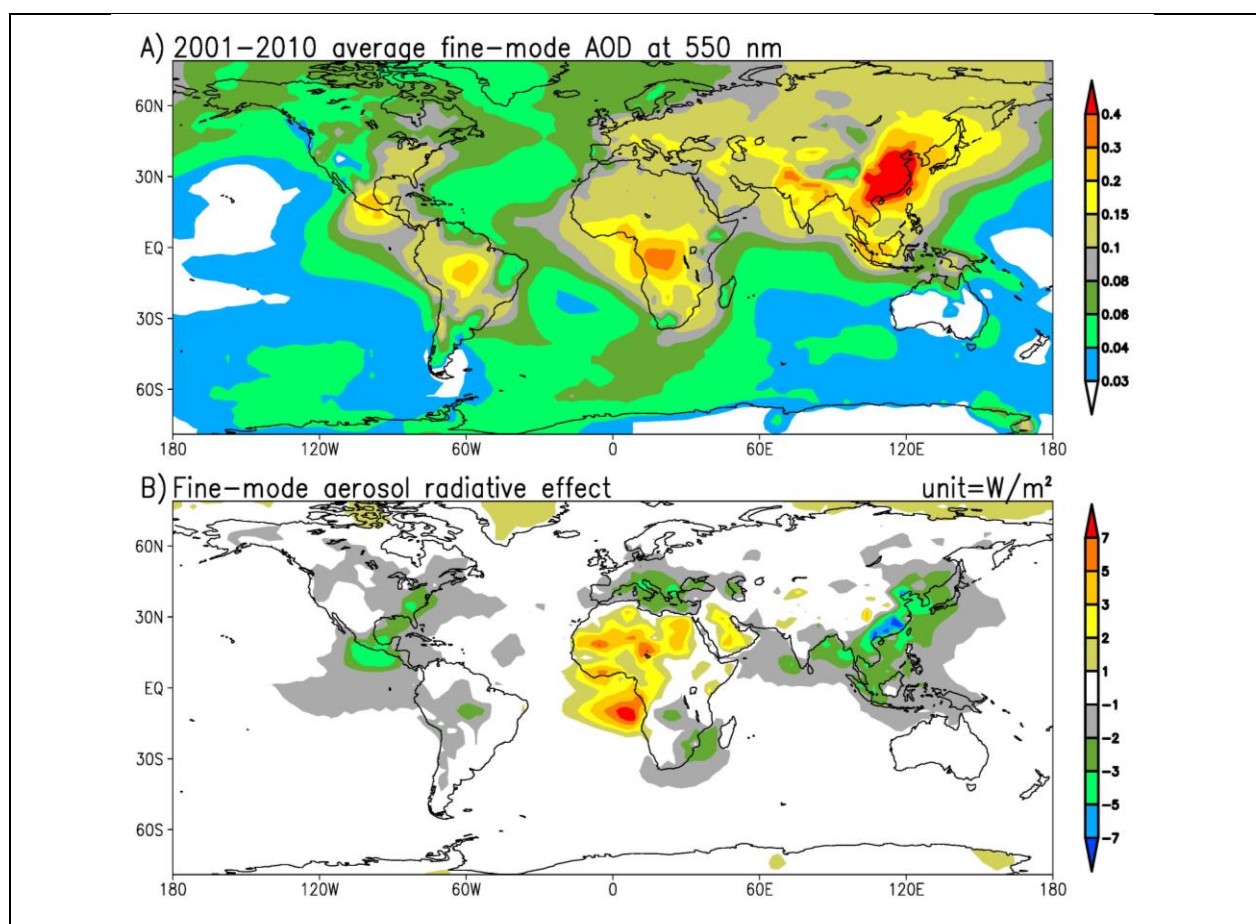

**Figure 3.** A) 2001-2010 mean 550 nm fine-mode AOD (fAOD) obtained by integrating MODIS, MISR, and AERONET data. B) 2001-2010 mean direct fine-mode aerosol radiative effect at TOA in units of Wm$^{-2}$; this estimate includes natural fine-mode particles.


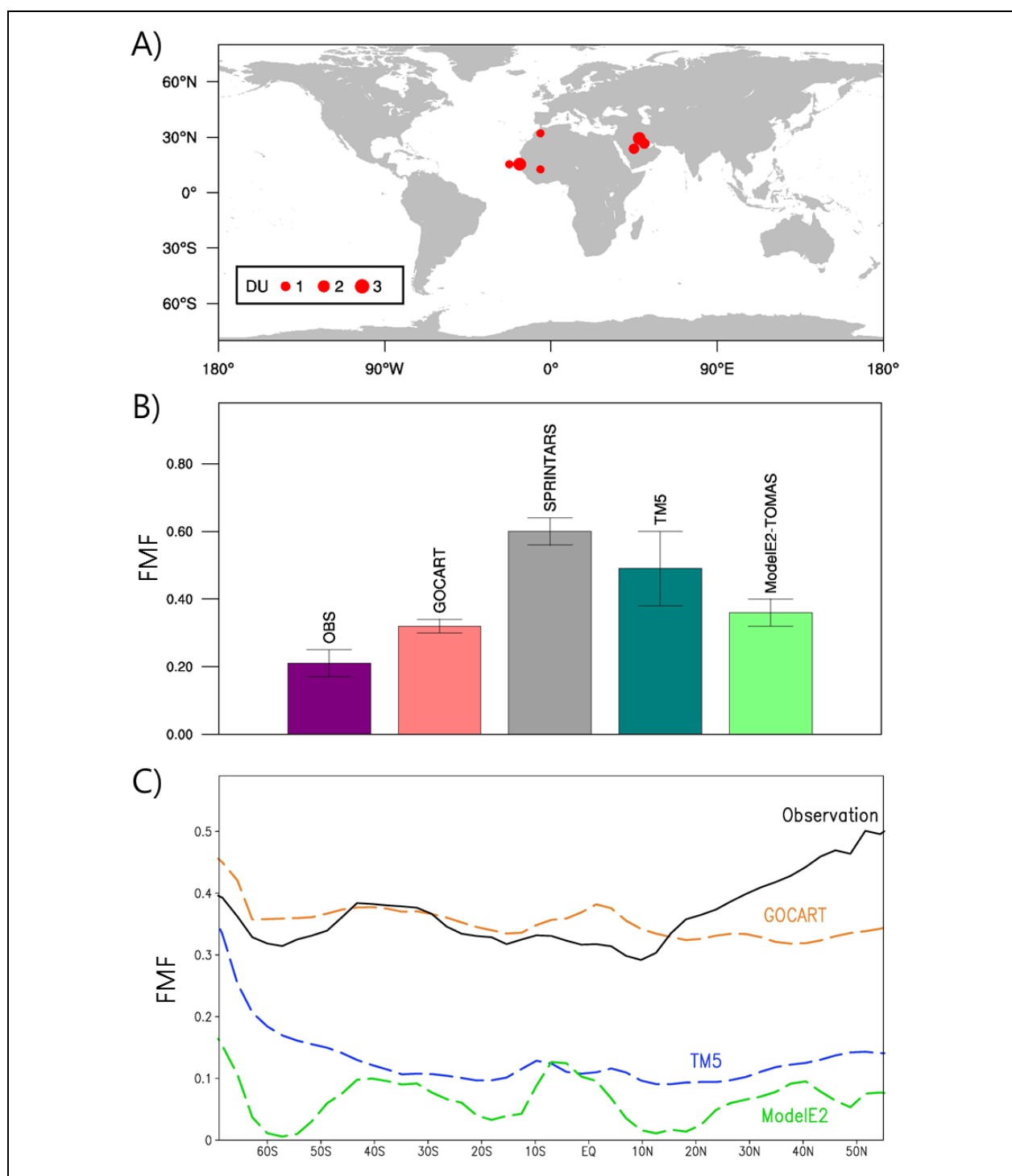

**Figure 4.** Comparison of simulated and observed fine-mode fraction (FMF) at 550 nm. A) Chosen dust-dominated (DU: red dots) AERONET sites. The dot size is proportional to the number of AERONET data from decadal means (2001-2010) for each calendar month. B) Simulated and observed FMF averaged over the chosen dust dominated sites. FMF averages are made by the average AOD and fAOD. The uncertainty represents ±1.0 standard deviation resulting from variation over the sites. FMFs here include the contribution from non-dust

particles. C) Sea salt AOD FMF along the 180th meridian (180° longitude), using annual average AODs. For observation, total FMF (instead of sea salt AOD FMF) is displayed.



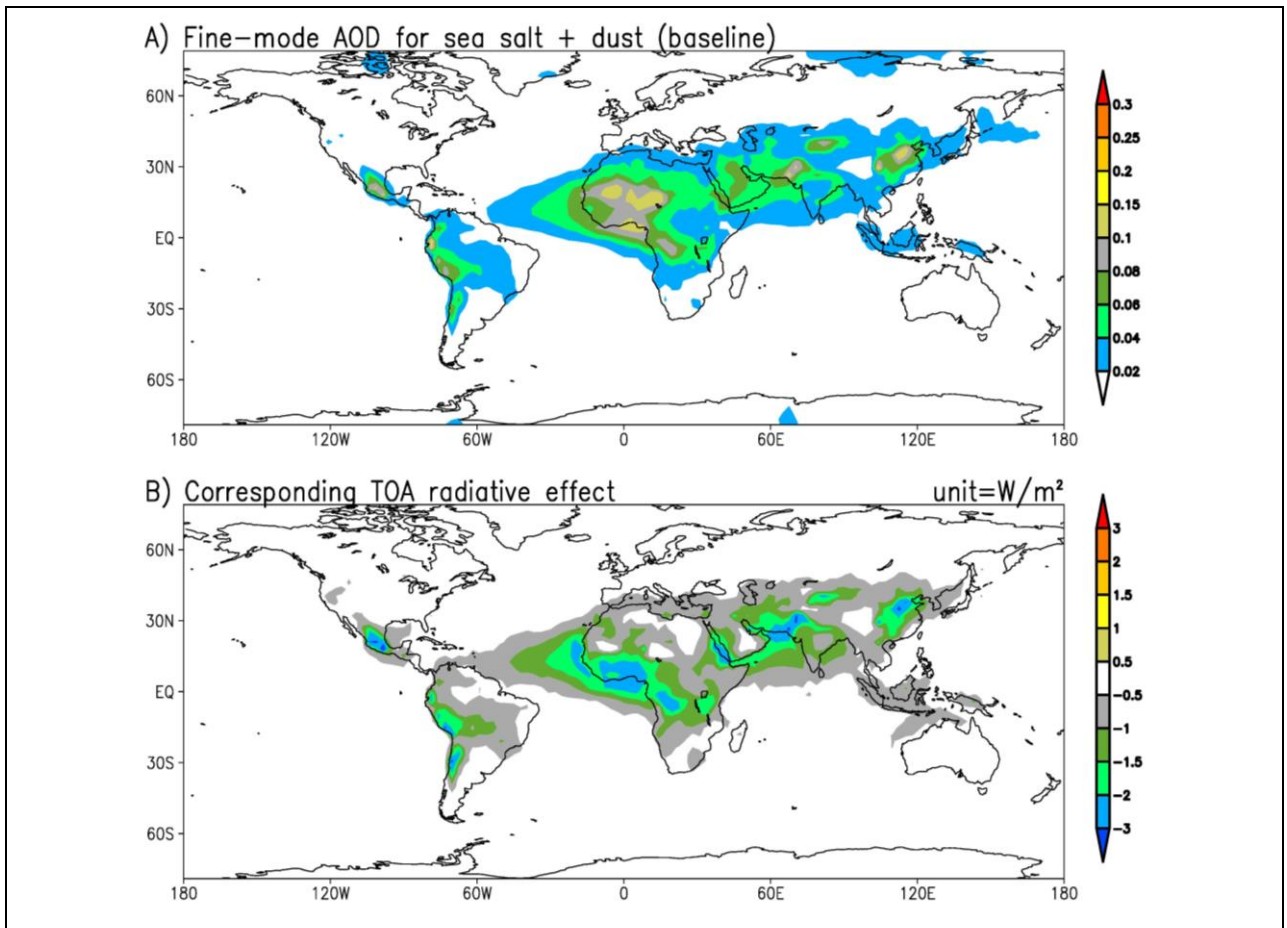

**Figure 5.** A) 2001-2010 mean fine-mode AOD at 550 nm for sea salt and dust, which is calculated as simulated

ratio × observational large-mode AOD, where the simulated ratio refers to

$\frac{\text{Model−mix fine−mode AOD for sea salt and dust}}{\text{Model−mix large−mode AOD for sea salt and dust}}$. The observational large-mode AOD is computed by integrating

AERONET, MODIS and MISR data. Model mix is an optimal mixture of GOCART, TM5 and ModelE2-TOMOS

AOD simulations. B) Aerosol direct radiative effect due to the sea salt and dust fine mode aerosols.

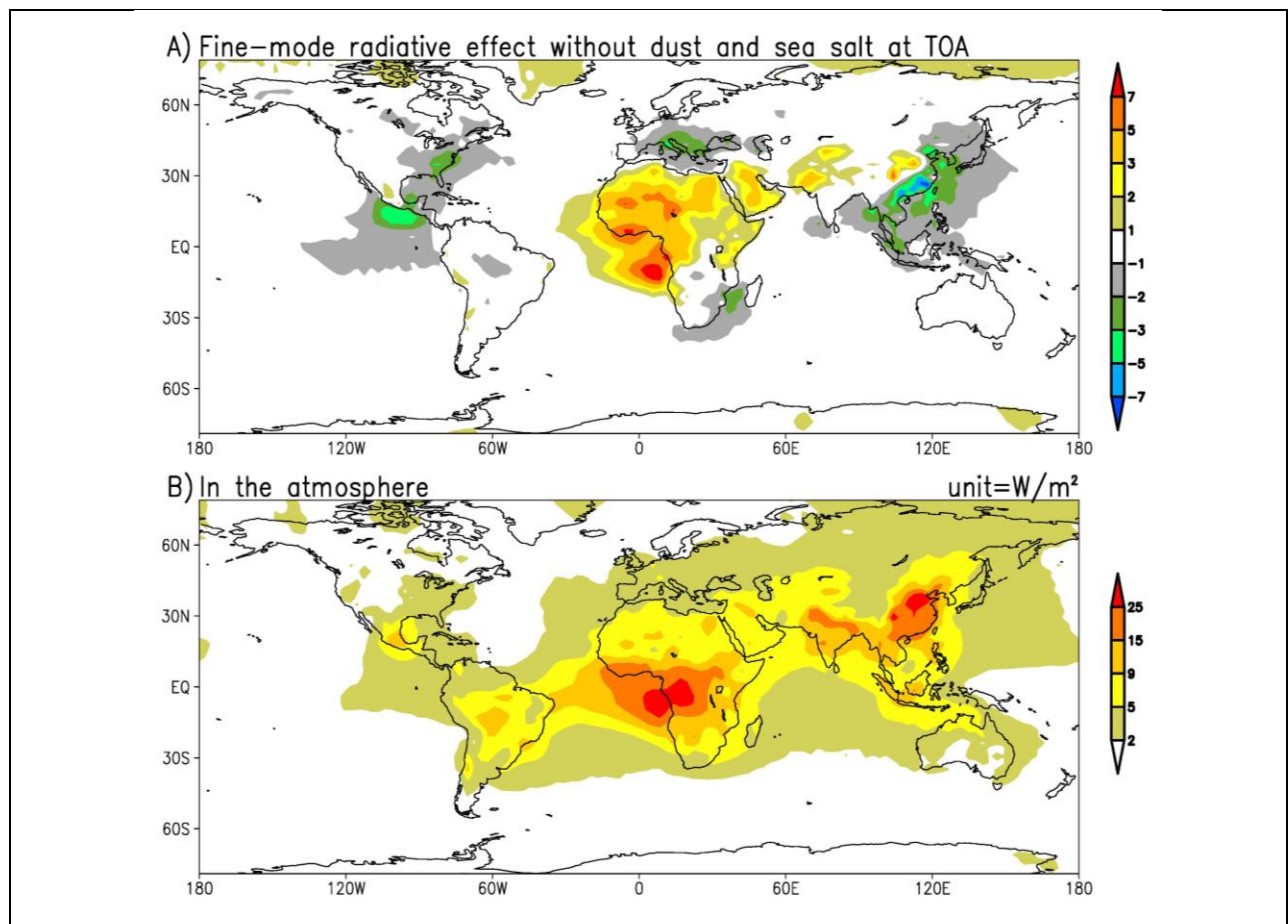

**Figure 6.** Direct fine-mode aerosol radiative effect without dust and sea salt in units of Wm$^{-2}$ (baseline). 2001-2010 mean values.








