# Peer review of "Global fine-mode aerosol radiative effect, as constrained by 1 comprehensive observations 2"

_Atmospheric Chemistry and Physics, 2016_

## Referee Comment (RC1) · Anonymous Referee #1 · 8 Feb 2016

Quantifying the direct radiative forcing of anthropogenic aerosols can be attempted either using observational or model data, or some combination of both. While models can cleanly separate anthropogenic and natural aerosols, aerosol simulations are still subject to large uncertainty. The use of observations alone is also problematic, due to issues regarding spatial coverage (AERONET) and/or accuracy (satellite observations of aerosols over land), and furthermore, assumptions are needed for separating the anthropogenic and natural aerosols. The current paper aims at providing the "most observational estimate of direct aerosol radiative forcing to date". For this end, available aerosol observations from AERONET, MISR and MODIS, along with vertical profile information from CALIPSO, are combined with GOCART simulations (and other model

results in Sect. 5). The main outcome of the paper is a best estimate of $-0.10$ W m$^{-2}$ for the global mean TOA direct radiative forcing of anthropogenic aerosols, with a quoted uncertainty range of $-0.28$ to $+0.05$ W m$^{-2}$.

I find this paper an interesting contribution worth publishing in ACP. However, while the paper is concise, it is currently not well-written, and it takes the reader (or reviewer) too much effort to understand, in detail, what you actually did and why. The clarity can and must be improved before the paper can be accepted for ACP. Specific suggestions are given both in the major comments (comment 3) and minor comments.

**Major comments**

1. Some assumptions in this paper need more discussion. In particular, to separate the anthropogenic direct aerosol forcing from the total aerosol direct radiative effect, it is assumed that (i) fine-mode aerosols are anthropogenic, except for fine-mode sea-salt and dust aerosols, and that (ii) all coarse-mode aerosols are natural. Neither of these assumptions is strictly true. First, the fine mode also includes natural particles (e.g., biogenic SOA from natural non-combustion sources, and BC from combustion, e.g. naturally occurring forest fires). Note that Bond et al. (2013) give (in their Sect 10.4.1 and Fig. 35) a best estimate of $+0.17$ W m$^{-2}$ for the preindustrial (year 1750) direct radiative effect of BC (as compared with a total natural+anthropogenic direct effect of $+0.88$ W m$^{-2}$ in year 2005). Second, some of the coarse-mode aerosols are anthropogenic. In particular, note that the IPCC AR5 best estimate of the direct radiative forcing (or "aerosol-radiation interaction") of $-0.35$ W m$^{-2}$ includes a contribution of $-0.1$ W m$^{-2}$ from changed dust emissions due to human activities (though with a large uncertainty range, from $-0.3$ to $+0.1$ W m$^{-2}$). See Table 8.4 in Myhre et al. (2013).

Separating out natural BC or anthropogenic dust would most likely be very difficult in the current framework, and I'm not suggesting that this be attempted. However, these issues should be discussed. If these factors were considered, it would presumably make your best estimate of the direct radiative forcing of anthropogenic aerosols more negative, and more in line with IPCC AR5.

REFERENCES:

*Bond, T. et al., 2013: Bounding the role of black carbon in the climate system: A scientific assessment. J. Geophys. Res., 118, 5380–5552.*

*Myhre, G., D. Shindell, et al. 2013: Anthropogenic and natural radiative forcing. In: Climate Change 2013: The Physical Science Basis. Contribution of Working Group 1 to the Fifth Assessment Report of the Intergovermental Panel of Climate Change.*

2. In section 5, the contribution of dust and sea salt to the fine-mode AOD (hereafter: $\mathrm{AOD_{SD,fine}}$) is estimated as

$$\mathrm{AOD_{SD,fine}} = \text{coarse-mode AOD} \times \frac{\mathrm{SD\_FMF}}{1 - \mathrm{SD\_FMF}}, \tag{1}$$

where $\mathrm{SD\_FMF}$ refers to the simulated fine-mode fraction of sea salt + dust AOD. While I can understand the reasoning behind this formula, the estimate is sensitive to errors in $\mathrm{SD\_FMF}$. Especially when $\mathrm{SD\_FMF}$ is too large (i.e., too much of simulated dust + sea salt is in the fine mode and too little in the coarse mode) $\mathrm{AOD_{SD,fine}}$ can become overly large due to the factor $1 - \mathrm{SD\_FMF}$ in the denominator. In fact, nothing prevents the derived $\mathrm{AOD_{SD,fine}}$ from exceeding the total fine-mode AOD, or even the total AOD in the observations! This seems to happen over Peru, where the total AOD in Fig. 1A is below 0.2, and the fine-mode AOD in Fig. 3A is around 0.10–0.15, but the fine-mode

[Figure]

AOD for sea salt + dust in Fig. 5A exceeds locally 0.3. This is related to a large positive anthropogenic fine-mode TOA forcing in Fig. 6A.

At minimum, it should be ensured that the derived $\mathrm{AOD_{SD,fine}}$ does not exceed the observed total fine-mode AOD. A stricter upper limit for $\mathrm{AOD_{SD,fine}}$ might be derived by estimating some lower limit for the (absolute or fractional) contribution that other species than dust + sea salt make to the fine-mode AOD.

3. After reading Section 2 several times, I am still not quite sure how the aerosol optical properties were derived in this study, but I try to summarize my understanding here:

- AOD at 550 nm was derived by nudging MODIS and MISR data towards AERONET observations. Remaining data gaps were filled by a GOCART simulation.

- Single-scattering albedo and asymmetry parameter at 550 nm were derived by nudging GOGART simulations towards AERONET data.

- Fine-mode AOD at 550 nm was obtained by subtracting coarse-mode AOD at 500 nm from the AOD at 550 nm, subject to the assumption that the coarse-mode AOD is equal at 500 and 550 nm. However, it is not clear from Section 2.1 how the coarse-mode AOD was obtained. Which of AERONET, MODIS, MISR and GOCART were used (and how)?

- It is not quite clear how the fine-mode single-scattering albedo and asymmetry parameter at 550 nm were derived. But I suppose that for consistency, they were derived by subtracting the corresponding coarse-mode values (with appropriate weighting by AOD) from the total single-scattering albedo and asymmetry parameter, and that the coarse-mode single-scattering albedo and asymmetry parameter were prescribed based on GOCART as explained in Sect 2.3.
[Figure]

- It is stated that the spectral dependence of single-scattering albedo is treated using an coalbedo Ångström exponent CA_AE, which is derived by nudging the GOCART simulation towards AERONET data. However, nothing is said about the spectral dependence of asymmetry parameter. Was it assumed spectrally independent? More importantly, it is not clear how the spectral dependence of AOD was treated. Section 2.3 suggests that you assumed an Ångström exponent of 0 for coarse-mode (dust and sea-salt) and 1.85 for the fine-mode dust and sea-salt (but nothing said about other fine-mode species). Did you use any observations to constrain the Ångström exponent?

If this interpretation is correct, you should make an effort to rewrite Sect. 2 so that it is easy for the reader to get this information (currently it is not). If not, the need for clearer writing is even more acute.

**Minor comments**

1. p. 1, 8th line of abstract (and elsewhere). In IPCC reports, "forcing" refers to changes in the radiation budget, so it would be better to reserve the term "radiative forcing" to the effect of anthropogenic aerosols. Thus, when discussing the impact of all (natural + anthropogenic) or natural aerosols on the radiative budget, "aerosol radiative effect" is preferable.

2. p. 1, first paragraph of Introduction: I assume that what you consider here is the climatic effect of anthropogenic aerosols. Please state this explicitly.

3. On p. 1 and 2, there are several citations to (Myhre, 2013), while it should be (Myhre et al., 2013).

4. On p. 2: The concepts of "fine-mode" and "coarse-mode" (or "large-mode") aerosols are central for this study, but their meaning is not explained adequately. E.g., can the modes be defined in terms of a size limit?

5. p. 3, first paragraph of Sect. 2.2: mention also the asymmetry parameter.

6. p. 3: There seems to be something wrong with the equation for $N\_ASY_j$. In case that there is no difference between the AERONET and GOCART values, $N\_ASY_j$ becomes 0!

7. p. 3: If the locations for AERONET stations are indexed with $i$ and model grid boxs with $j$, why are there two indices $(j, i)$ for both $AERONET_{j,i}$ and $G\_ASY_{j,i}$? Why is it not simply $AERONET_i$ and $G\_ASY_j$? The same goes for the following equation for $1 - N\_SSA_j$.

8. On p. 4, three different sets of aerosol optical properties are considered. While the selected parameters do not seem unreasonable, it is not clear how they were selected (no single reference is given!), nor whether the selected range represents well the real uncertainty range. For example, the single-scattering albedo of BC is taken here to be either 0.14 or 0.19. For comparison, Bond et al. (JGR 2013) cite in their Section 3.8.3 measured values between 0.10 and 0.28, and modelled values for aggregates of pure BC particles of 0.1 to 0.3. Further, a value of 0.36 based on Mie theory is applied in Appendix 3.1. The authors should either justify their values and/or test the impact of a wider parameter range.

9. p. 4. Please consider the use of a more rigorous scientific notation in your equations. Now, for example, "dust" refers to the total AOD for dust in the equations for SSA1, SSA2, and SSA3, the scattering AOD for dust in the equation for ASY, and presumably again the total AOD for dust in the equation for CA_AE. Furthermore, "BC"

refers to the GOCART simulated BC AOD in the equations for SSA1 and SSA3, but twice this value in the equation for SSA2. So apparently "total_AOD" is also a slightly different beast in the equations for SSA1 and SSA3 than in the equation for SSA2. Honestly, this becomes rather confusing. Similarly, on p. 3, it would be clearer to use AERONET_ASY in the first equation and AERONET_SSA in the second equation.

10. p. 4. Also, please consider using equation numbers to make it easier to refer to the equations. (Indeed, they would have made the previous comment shorter!).

11. p. 4. In the equation for CA_AE, "CA" refers to both "coalbedo" (on the left) and to "carbonaeous aerosol" on the right. To avoid confusion, I suggest the use of BC+OC (or, preferably, BC_AOD + OC_AOD) on the right hand side.

12. p. 4–5. Is there a reference for the equations for CA_AE in Sect. 2.2 and 2.3?

13. p. 4. While it is probably true that the uncertainty in asymmetry parameter is unimportant compared to the uncertainty in SSA, this uncertainty cannot be quantified by just doubling the BC AOD (BC makes only a minor contribution to the total AOD, so it has a small impact on the average asymmetry parameter). A more reasonable estimate might be obtained by perturbing the total aerosol asymmetry parameter by, say, $\pm 0.05$. Obviously, even this would be a subjective choice, but since all your asymmetry parameter values are rounded to nearest 0.05, it seems reasonable to assume that the true values are not known much more precisely than that.

14. p. 4. "Global simulations" is a somewhat misleading title for Section 2.3. No global simulations are described here, just the assumptions about aerosol optical properties.

15. p. 5, 8th line of Sect. 2.4. Is "vertically summed aerosol extinction" the same as AOD, or something else?

16. p. 5–6. Section 3 should contain more information about the input data for the MACR model. Currently, it describes some details related to low clouds but does not explain how, in the first place, clouds were treated and where the cloud data came from. I realize that this information might be found in Choi and Chung (2014), but the main points should be repeated here to make this paper stand alone.

17. p. 8, first line: The criteria for selecting dust-dominated Aeronet sites are not intuitively clear. For example, why is it required that the absorption AOD exceeds 0.03? Dust is relatively weakly absorbing, with a single-scattering albedo of 0.96 at 550 nm used in this paper.

18. p. 9, last paragraph: It is concluded that the direct radiative forcing of anthropogenic aerosols is close to zero. Another important result, which would deserve some more discussion, is that the anthropogenic aerosols strongly increase atmospheric absorption (by 3.64 W m$^{-2}$, according to Table 1) and decrease the net solar radiation at the surface (by 3.74 W m$^{-2}$). This suggests that the primary climatic impact of anthropogenic aerosols is not that of cooling the planet but reducing precipitation (less solar energy available for evaporation, and less latent heating needed for balancing radiative cooling). Reduced precipitation due to absorbing aerosols has of course been discussed earlier, e.g. Ramanathan and Carmichael (2008) and Ming et al. (2010).

REFERENCES:

*Ramanathan, V. and G. Carmichael, 2008: Global and regional climate changes due to black carbon. Nature Geosci., 1, 221-227.*

*Ming, Y., V. Ramaswamy and G. Persaad, 2010: Two opposing effects of absorb-*

*ing aerosols on global mean precipitation. Geophys. Res. Lett., 37, L13701, doi:10.1029/2010GL042895.*

19. p. 10: Where was the review paper by Chung (2012) published?

20. p. 16: Fig. 4. The dot size scale (1, 2, 3) related to the number of AERONET climatological data is uninformative. Please be more explicit.

---

## Author Comment (AC1) · 9 Feb 2016

We apologize that the paper was not clear enough to you. We did not claim that the global mean TOA direct radiative forcing of anthropogenic aerosols is ($-0.28 \sim +0.05$) Wm-2. This number is for total (anthropogenic + natural) carbonaceous, sulfate and nitrate aerosols. Furthermore, we did not attempt to estimate the anthropogenic fraction of these carbonaceous, sulfate and nitrate aerosols. In other words, we did not assume that fine-mode aerosols are anthropogenic except for fine-mode sea-salt and dust aerosols. However, we did assume that all coarse-mode aerosols are natural, which, as the reviewer pointed out, is not very true due to dust changes.

We will revise the manuscript to improve the clarity and address other issues. Thanks

for the valuable feedbacks.

---

## Referee Comment (RC2) · Chul E. Chung et al. · 1 Mar 2016

In this paper, "Global direct aerosol radiative forcing, as constrained by comprehensive observations", the authors use a suite of observations from satellite and in-situ platforms to determine the optical properties of aerosol, from which the direct radiative effect can be determined when incorporated with a radiative transfer model and some information from aerosol models. The authors use the fine mode aerosol without sea salt and dust as a proxy for anthropogenic aerosol, and find that the radiative effect is more positive that previous estimates, stating that it is close to zero.

Main points

One key issue throughout the paper is the distinction between direct radiaitve forcing (DRF) and direct radiative effect (DRE). Generally, aerosol radiative forcing is used

when talking about the change in radiative effect relative to pre-industrial levels. Here the analysis is concerned with the direct radiative effect, no comparison to the pre-industrial aerosol radiative effect is made (other than in the final section). For clarity, this needs correcting throughout the paper to only use radiative forcing where the anthropogenic component is being considered, and care also taken to make sure comparisons are not being made with numbers from other literature that is concerned with the aerosol radiative forcing. Furthermore, if the authors intend to consider the anthropogenic aerosol radiative forcing then I think a more detailed consideration of the pre-industrial aerosol composition is necessary. Otherwise the result of this paper is really the fine mode DRE from non-dust and sea salt aerosols, not the anthropogenic DRF.

My second main point relates to the attribution of the fine mode aerosol to the BC, OC and inorganic species...

It seems strange that the fine-mode aerosol radiative effect quite abruptly ends at the west coast of Africa (Figure 3b). If this estimate includes natural aerosol, as stated, then I would expect a contribution from fine dust aerosol over the mid-Atlantic. Does the radiative effect from fine dust really drop away that fast? Figure 5b suggests that there is a significant negative effect in the mid-Atlantic from dust and sea-salt, is this really balanced by a positive impact of other aerosol in this region?

Related to the above, it is not clear from Figure 3 if this is surface or TOA, so perhaps I'm conflating two different metrics here.

More importantly, the authors allude to the strange positive forcing from the 'anthropogenic' fine aerosol over the Sahara being the result of northward transport of biomass burning aerosol and bright desert surfaces. This is a testable hypothesis: there should be a relatively clear seasonal cycle in the impact, with most biomass burning emissions in the boreal winter time in West Africa. Is this the case?

Also, do the regions of highest radiative effect in Africa in Figure 6a align with the

brightest regions of the surface reflectance assumed? The regions of high DRE do align with the locations of high AOD in Figure 1A that are generally considered the result of strong dust sources at those locations. There is potential for a similar problem in the Middle East and the Asian deserts too (it looks like there is non-dust fine mode AOD in the Teklemakan desert, which would be strange). This might indicate mis-attribution of dust aerosol as non-dust fine mode aerosol. The authors make efforts to consider the uncertainties involved in the estimate, which is appreciated; however, it is not clear how well the sensitivity studies would account for this likely mis-attribution. In this case, any error would cause a more positive radiative effect from the fine mode aerosol. This is a key point of the paper, so I think it is important to understand any potential bias.

Smaller points

I think the language in the final sentence of the abstract is too strong. Consider replacing "near-zero" with "closer to zero than previous estimates".

In the final paragraph of the first section I think it is important that the authors state when they use model information in this study. Based on that description it seems like models are not used, but this is not really true. An expanded discussion here would allow a better description of how this study builds upon the other studies cited, related to the minimization of reliance on models.

The single sentence after the Section 2 heading either needs expanding or removing.

Do you have some metric of the nudging that is applied to MODIS and MISR that can be stated in the paper? The change in the slope perhaps?

With the 'nudging' of the GOCART model to the AERONET observations, in regions for which there are no AERONET observations is the result essentially just the model? Were any other functional forms than the $1/d^4$ used? That suggests a very rapid relaxation back to the model values. Do you have any maps or numbers to indicate

how strongly this nudging impacts the SSA and the ASY?

The equations on pg4 are not very clear, consider using an equation editor and symbols instead of simply writing as text.

The high FMF from models at dust sites, relative to observations (Figure 4b), may be an issue with dust emissions favoring too small sizes (Kok et al., 2011). The authors should probably mention this, unless those models have been updated to reflect the latest emission distributions.

At the end of the first paragraph in section 6, the way the numbers are displayed is unnecessarily confusing. I think the final range can be quoted alone.

Section 6: "At least > -0.28Wm-2" please reword, e.g. "more positive than -0.28Wm-2", "unlikely to be less than -0.28Wm-2"

I'm not sure about the final sentence of the paper. There is still some reliance on models, albeit less than previous studies. Please consider revising the closing remarks.

Table 1 - break the first column into two so that the property e.g. fine-mode DRE, can be displayed in the first column and then the specifics of that case put in the second column for ease of reading.

Fig 1b and Fig 3b - state whether surface or TOA Fig 1b says DRE correctly, but then Fig 3b is called the aerosol forcing, please correct this throughout the paper Fig 6 - would it be useful to show the surface fine mode DRE here as well, for completeness?

Bellouin et al. (2008) reference has n/a errors for page numbers

References

Kok, J. F.: A scaling theory for the size distribution of emitted dust aerosols suggests climate models underestimate the size of the global dust cycle, PNAS, 108(3), 1016–1021, doi:10.1073/pnas.1014798108, 2011.

---

## Author Comment (AC2) · 2 Mar 2016

We thank the reviewer for many constructive suggestions. We will revise the paper accordingly.

We want to clarify one point. We did not use the fine mode aerosols without sea salt and dust as a proxy for anthropogenic aerosol, as the reviewer thought. We apologize for this confusion and the other reviewer had the same confusion. We will make sure that we'll remove this confusion during the revision.

The reviewer is right that our analysis was concerned with the direct radiative effect. We did not estimate direct radiative forcing but speculated it based on our estimate of

direct radiative effect. We will make this super clear during the revision.

---

## Referee Comment (RC3) · Anonymous Referee #1 · 4 Mar 2016

Thank you for the clarification. Indeed, it is stated in the last sentence of your abstract that your estimate does not exactly represent the radiative forcing of anthropogenic aerosols. However, it was easy to misinterpret this, since (i) the concepts of aerosol radiative effect and (anthropogenic) radiative forcing are not properly separated in the paper; (ii) clearly, the underlying motivation of the paper seems to be to estimate the direct radiative forcing due to anthropogenic aerosols (you just cannot quite get there) and (iii) your estimate is compared in the abstract with the value $-0.35 \pm 0.5$ W m$^{-2}$, which is the IPCC AR5 estimate for the direct radiative forcing (or "aerosol-radiation interaction") of anthropogenic aerosols. Regarding this latter point, it is worth noting that your best estimate of $-0.10$ W m$^{-2}$ for the radiative effect of carbonaceous, sulfate

and nitrate aerosols actually seems to be quite consistent with the IPCC AR5 estimate of the anthropogenic radiative forcing. To get from your estimate to the anthropogenic forcing, one should (at least):

- Add the contribution of "anthropogenic" dust. Assume, for the sake of the argument, that the IPCC AR5 best estimate ($-0.10$ W m$^{-2}$) is valid.

- Subtract the contribution of natural BC. Assume that this is represented by the Bond et al. (2013; Sect. 10.4.1 and Fig. 35) year 1750 value of $+0.17$ W m$^{-2}$.

Thus, one ends up at: $-0.10$ W m$^{-2}$ $-0.10$ W m$^{-2}$ $-0.17$ W m$^{-2}$ = $-0.37$ W m$^{-2}$, which happens to be very close to $-0.35$ W m$^{-2}$. Therefore I would argue that the statement on p. 9, line 4 ("we posit that the aerosol direct radiative forcing is less negative than the concensus") is not necessarily true.

---

## Author Comment (AC3) · 4 Mar 2016

Thanks for better understanding our paper. You are quite right about your calculations (i.e., adding anthropogenic dust and subtracting natural BC). We also need to further subtract the contributions from natural sulfate, natural organic aerosol and natural nitrate. These aerosols give a cooling and so subtracting these aerosols make the net forcing less negative. This is why we said that the forcing is less negative than the consensus. We will make this clearer during the revision.

---

## Author Response (AR1)

Referee #1
Quantifying the direct radiative forcing of anthropogenic aerosols can be attempted either using observational or model data, or some combination of both. While models can cleanly separate anthropogenic and natural aerosols, aerosol simulations are still subject to large uncertainty. The use of observations alone is also problematic, due to issues regarding spatial coverage (AERONET) and/or accuracy (satellite observations of aerosols over land), and furthermore, assumptions are needed for separating the anthropogenic and natural aerosols. The current paper aims at providing the "most observational estimate of direct aerosol radiative forcing to date". For this end, available aerosol observations from AERONET, MISR and MODIS, along with vertical profile information from CALIPSO, are combined with GOCART simulations (and other model results in Sect. 5). The main outcome of the paper is a best estimate of $-0.10$ W m$^{-2}$ for the global mean TOA direct radiative forcing of anthropogenic aerosols, with a quoted uncertainty range of $-0.28$ to $+0.05$ W m$^{-2}$ .

➔ We apologize that the paper was not clear enough to the reviewer. We did not claim that the global mean TOA direct radiative forcing of anthropogenic aerosols is ($-0.28 \sim +0.05$) Wm$^{-2}$. We and the reviewer already exchanged texts on this at the interactive website. During the revision, we modified terms and texts to improve the clarity. In addition, we changed the main aim of the paper from "most observational estimate of direct aerosol radiative forcing to date" to "most observational estimates of direct fine-mode aerosol radiative effect". The revised title reflects this.

I find this paper an interesting contribution worth publishing in ACP. However, while the paper is concise, it is currently not well-written, and it takes the reader (or reviewer) too much effort to understand, in detail, what you actually did and why. The clarity can and must be improved before the paper can be accepted for ACP. Specific suggestions are given both in the major comments (comment 3) and minor comments.

➔ We thank the reviewer for finding our paper interesting. We attempted to improve the clarity during the revision. Some of these revisions are explained below in response to some of the comments.

Major comments
1. Some assumptions in this paper need more discussion. In particular, to separate the anthropogenic direct aerosol forcing from the total aerosol direct radiative effect, it is assumed that (i) fine-mode aerosols are anthropogenic, except for fine-mode sea-salt and dust aerosols, and that (ii) all coarse-mode aerosols are natural. Neither of these assumptions is strictly true. First, the fine mode also includes natural particles (e.g., biogenic SOA from natural non-combustion sources, and BC from combustion, e.g. naturally occurring forest fires). Note that Bond et al. (2013) give (in their Sect 10.4.1 and Fig. 35) a best estimate of $+0.17$ W m$-2$ for the preindustrial (year 1750) direct radiative effect of BC (as compared with a total natural+anthropogenic direct effect of $+0.88$ W m$-2$ in year 2005). Second, some of the coarse-mode aerosols are anthropogenic. In particular, note that the IPCC AR5 best estimate of the direct radiative forcing (or "aerosol-radiation interaction") of $-0.35$ W m$^{-2}$ includes a contribution of $-0.1$ W m$^{-2}$ from changed dust emissions due to human activities (though with a large uncertainty range, from $-0.3$ to $+0.1$ W m$^{-2}$ ). See Table 8.4 in Myhre et al. (2013).

Separating out natural BC or anthropogenic dust would most likely be very difficult in the current framework, and I'm not suggesting that this be attempted. However, these issues should be discussed. If these factors were considered, it would presumably make your best estimate of the direct radiative forcing of anthropogenic aerosols more negative, and more in line with IPCC AR5.

➔ Again, we did not assume that fine-mode aerosols are anthropogenic except for fine-mode sea-salt and dust aerosols. However, we did assume that all coarse-mode aerosols are natural. We thank the reviewer for pointing out that the IPCC estimate of aerosol direct forcing includes a dust forcing. To reduce the confusion, we changed the title of the paper and terms throughout the paper. We also re-wrote much of section 6 to accurately compare our observation estimate to the IPCC estimate. See "The global aerosol direct radiative forcing estimate in the 5th IPCC report is -0.35 Wm$^{-2}$ (Myhre et al., 2013), and this includes a dust forcing of -0.10 Wm$^{-2}$. Thus, the IPCC estimate is that anthropogenic carbonaceous, sulfate and nitrate aerosols pose a radiative forcing of -0.25 Wm$^{-2}$, while our observational estimate of total (anthropogenic + natural) carbonaceous, sulfate and nitrate aerosol forcing is -0.11 Wm$^{-2}$. The anthropogenic fraction (or pre-industrial fraction) of carbonaceous, sulfate and nitrate aerosols is uncertain. Black carbon, the only warming aerosol species in carbonaceous aerosol (black carbon + organic aerosol), sulfate and nitrate aerosol is known to be more anthropogenic than organic aerosols are (Bond et al., 2011). If the anthropogenic fraction of black carbon is similar to that of nitrate and sulfate aerosol, the aerosol direct radiative forcing becomes < -0.11 Wm$^{-2}$ in our observational estimation, which means that aerosol direct forcing is less negative than the consensus as expressed in the 5th IPCC report."

2. In section 5, the contribution of dust and sea salt to the fine-mode AOD (hereafter: $AOD_{SD,fine}$) is estimated as

$AOD_{SD,fine}$ = coarse-mode AOD × SD_FMF / (1 − SD_FMF) ,    (1)

where SD_FMF refers to the simulated fine-mode fraction of sea salt + dust AOD. While I can understand the reasoning behind this formula, the estimate is sensitive to errors in SD_FMF. Especially when SD_FMF is too large (i.e., too much of simulated dust + sea salt is in the fine mode and too little in the coarse mode) $AOD_{SD,fine}$ can become overly large due to the factor 1 − SD_FMF in the denominator. In fact, nothing prevents the derived $AOD_{SD,fine}$ from exceeding the total fine-mode AOD, or even the total AOD in the observations! This seems to happen over Peru, where the total AOD in Fig. 1A is below 0.2, and the fine-mode AOD in Fig. 3A is around 0.10–0.15, but the fine-mode AOD for sea salt + dust in Fig. 5A exceeds locally 0.3. This is related to a large positive anthropogenic fine-mode TOA forcing in Fig. 6A.

At minimum, it should be ensured that the derived $AOD_{SD,fine}$ does not exceed the observed total fine-mode AOD. A stricter upper limit for $AOD_{SD,fine}$ might be derived by estimating some lower limit for the (absolute or fractional) contribution that other species than dust + sea salt make to the fine-mode AOD.

➔ We thank the reviewer so much for this constructive suggestion! The reviewer is right that $AOD_{SD,fine}$ does exceed the total fine-mode AOD on rare occasions. During the revision, we limited the $AOD_{SD,fine}$ to be < 99% of the total fine-mode AOD, which resulted in modified Table 1, Fig. 5 and Fig. 6. We find that this modification primarily modifies $AOD_{SD,fine}$ over Peru and changes the global average forcing very insignificantly (by less than 0.01 Wm$^{-2}$). We added the following sentences in the manuscript: "On rare occasions, $\frac{SD\_FMF}{1-SD\_FMF}$ becomes unrealistically

large. To prevent this, we limit fine-mode sea salt and dust AOD to be < 99% of total fine-mode AOD."

3. After reading Section 2 several times, I am still not quite sure how the aerosol optical properties were derived in this study, but I try to summarize my understanding here:
• AOD at 550 nm was derived by nudging MODIS and MISR data towards AERONET observations. Remaining data gaps were filled by a GOCART simulation.
• Single-scattering albedo and asymmetry parameter at 550 nm were derived by nudging GOGART simulations towards AERONET data.
• Fine-mode AOD at 550 nm was obtained by subtracting coarse-mode AOD at 500 nm from the AOD at 550 nm, subject to the assumption that the coarse-mode AOD is equal at 500 and 550 nm. However, it is not clear from Section 2.1 how the coarse-mode AOD was obtained. Which of AERONET, MODIS, MISR and GOCART were used (and how)?
• It is not quite clear how the fine-mode single-scattering albedo and asymmetry parameter at 550 nm were derived. But I suppose that for consistency, they were derived by subtracting the corresponding coarse-mode values (with appropriate weighting by AOD) from the total single-scattering albedo and asymmetry parameter, and that the coarse-mode single-scattering albedo and asymmetry parameter were prescribed based on GOCART as explained in Sect 2.3.
➔ We modified section 2 to more clearly explain the dataset (e.g., coarse-mode AOD). However, we don't need to deal with SSA for fine-mode aerosols, since we computed fine-mode direct radiative effect as the difference between the coarse-mode and total (coarse + fine modes) aerosol radiative effect. This was mentioned in the old manuscript, but to make this clearer, we added the following sentences in section 2: "As for optical properties (for example, SSA) for fine-mode aerosols, we do not need to address them since we choose to compute fine-mode direct radiative effect as the difference between the coarse-mode and total (coarse + fine modes) aerosol radiative effect."

• It is stated that the spectral dependence of single-scattering albedo is treated using an coalbedo Ångström exponent CA_AE, which is derived by nudging the GOCART simulation towards AERONET data. However, nothing is said about the spectral dependence of asymmetry parameter. Was it assumed spectrally independent? More importantly, it is not clear how the spectral dependence of AOD was treated. Section 2.3 suggests that you assumed an Ångström exponent of 0 for coarse-mode (dust and sea-salt) and 1.85 for the fine-mode dust and sea-salt (but nothing said about other fine-mode species). Did you use any observations to constrain the Ångström exponent?
➔ We apologize for not explaining the spectral dependence of ASY. We added the following sentence: "The spectral dependence of ASY is addressed as in Chung et al. (2005)." The revised manuscript also explains the spectral dependence of AOD; see "Also, AOD Ångström exponent from 2001 to 2010 is derived by adjusting the satellite data towards AERONET data as in Lee and Chung (2013)."

If this interpretation is correct, you should make an effort to rewrite Sect. 2 so that it is easy for the reader to get this information (currently it is not). If not, the need for clearer writing is even more acute.
➔ Again, we improved section 2 quite a bit during the revision.

Minor comments

1. p. 1, 8th line of abstract (and elsewhere). In IPCC reports, "forcing" refers to changes in the radiation budget, so it would be better to reserve the term "radiative forcing" to the effect of anthropogenic aerosols. Thus, when discussing the impact of all (natural + anthropogenic) or natural aerosols on the radiative budget, "aerosol radiative effect" is preferable.

➔ As we explained earlier, we changed the terms throughout the paper so that "radiative forcing" is exclusively used for anthropogenic aerosols.

2. p. 1, first paragraph of Introduction: I assume that what you consider here is the climatic effect of anthropogenic aerosols. Please state this explicitly.

➔ We inserted "anthropogenic" here.

3. On p. 1 and 2, there are several citations to (Myhre, 2013), while it should be (Myhre et al., 2013).

➔ Corrected as suggested.

4. On p. 2: The concepts of "fine-mode" and "coarse-mode" (or "large-mode") aerosols are central for this study, but their meaning is not explained adequately. E.g., can the modes be defined in terms of a size limit?

➔ This is a good point. We added the following sentences in section 1: "Aerosols have different sizes, and typically follow a bimodal structure in terms of fine mode and coarse mode (Kim et al., 2007; Viskari et al., 2012). Fine-mode aerosols usually have submicron sizes in diameter and these small particles are mostly anthropogenic." We also added the following sentence in section 2: "Note that the definition of fine mode in the present study thus follows that by the AERONET Spectral Deconvolution Algorithm as in Lee and Chung (2013)." Please note that the AERONET SDR (Spectral Deconvolution Algorithm) calculation of fine mode does not specify a size cut-off value.

5. p. 3, first paragraph of Sect. 2.2: mention also the asymmetry parameter.

➔ We can see that the first para gives an impression that we are only dealing with SSA. To rectify this, we added the following sentence in the first para: "We apply a similar procedure to ASY (Asymmetry Parameter) and other aerosol optical properties."

6. p. 3: There seems to be something wrong with the equation for $N\_ASY_j$ . In case that there is no difference between the AERONET and GOCART values, $N\_ASY_j$ becomes 0!

➔ We apologize for this typo. The mistake was corrected during the revision. Thanks a lot!

7. p. 3: If the locations for AERONET stations are indexed with i and model grid boxs with j, why are there two indices (j, i) for both $AERONET_{j,i}$ and $G\_ASY_{j,i}$? Why is it not simply $AERONET_i$ and $G\_ASY_j$? The same goes for the following equation for $1 - N\_SSA_j$ .

➔ The reviewer is right that $AERONET_i$ suffices. After a careful review, we think that $G\_ASY_{j,i}$ should be $G\_ASY_i$. We made this correction during the revision.

8. On p. 4, three different sets of aerosol optical properties are considered. While the selected parameters do not seem unreasonable, it is not clear how they were selected (no single reference is given!), nor whether the selected range represents well the real uncertainty range. For example,

the single-scattering albedo of BC is taken here to be either 0.14 or 0.19. For comparison, Bond et al. (JGR 2013) cite in their Section 3.8.3 measured values between 0.10 and 0.28, and modelled values for aggregates of pure BC particles of 0.1 to 0.3. Further, a value of 0.36 based on Mie theory is applied in Appendix 3.1. The authors should either justify their values and/or test the impact of a wider parameter range.

➔ We added sentences and modified sentences in this para. See "We chose parameters (e.g., 0.19 for BC SSA) in the above three equations from various observational studies (e.g., Magi, 2009; Magi, 2011). Additionally, in SSA2 (more absorbing case), we doubled the magnitude of BC AOD, given a notion (e.g., Chung et al., 2012) that simulated BC is significantly underestimated. We use the above three sets of simulated SSA in order to produce an initial estimate of the uncertainty in simulated SSA. Then, we nudged the 3 sets of simulated SSA towards the same AERONET SSA, which gave 3 sets of semi-observational SSA. Finally, we computed the average, maximum and minimum SSAs from the three sets of SSA over each grid and each calendar month, and then re-generated three sets of SSA (average: baseline; maximum; and minimum; see Fig. 2). This re-generation increases the global-average SSA difference between the least absorbing and most absorbing cases. We do this re-generation in an attempt to fully bracket the simulated SSA uncertainty. The last procedure (i.e., re-generation) assures that the final three sets of SSA depend insignificantly on the initial estimate of the simulated SSA uncertainty." Please note that high BC SSA values (such as 0.36) are based on a spherical treatment of BC shapes and thus not very realistic.

9. p. 4. Please consider the use of a more rigorous scientific notation in your equations. Now, for example, "dust" refers to the total AOD for dust in the equations for SSA1, SSA2, and SSA3, the scattering AOD for dust in the equation for ASY, and presumably again the total AOD for dust in the equation for CA_AE. Furthermore, "BC" refers to the GOCART simulated BC AOD in the equations for SSA1 and SSA3, but twice this value in the equation for SSA2. So apparently "total_AOD" is also a slightly different beast in the equations for SSA1 and SSA3 than in the equation for SSA2. Honestly, this becomes rather confusing. Similarly, on p. 3, it would be clearer to use AERONET_ASY in the first equation and AERONET_SSA in the second equation.

➔ We improved the manuscript in this regard. For example, see "$AERONET\_ASY_i$".

10. p. 4. Also, please consider using equation numbers to make it easier to refer to the equations. (Indeed, they would have made the previous comment shorter!).

➔ We added numbers for the first two equations in section 2.2, and adjusted the text accordingly.

11. p. 4. In the equation for CA_AE, "CA" refers to both "coalbedo" (on the left) and to "carbonaeous aerosol" on the right. To avoid confusion, I suggest the use of BC+OC (or, preferably, BC_AOD + OC_AOD) on the right hand side.

➔ Thanks for pointing out this potential confusion. We chose to modify CA_AE into CAl_AE. For carbonaceous aerosols, we consistently use CA_AOD or CA_SAOD in the revised manuscript.

12. p. 4–5. Is there a reference for the equations for CA_AE in Sect. 2.2 and 2.3?

➔ We adopted these values from the AERONET data analysis. We added the following sentence: "The chosen parameters (e.g., 2.215) in the ASY and CAl_AE equations are from preliminary AERONET data analysis."

13. p. 4. While it is probably true that the uncertainty in asymmetry parameter is unimportant compared to the uncertainty in SSA, this uncertainty cannot be quantified by just doubling the BC AOD (BC makes only a minor contribution to the total AOD, so it has a small impact on the average asymmetry parameter). A more reasonable estimate might be obtained by perturbing the total aerosol asymmetry parameter by, say, ±0.05. Obviously, even this would be a subjective choice, but since all your asymmetry parameter values are rounded to nearest 0.05, it seems reasonable to assume that the true values are not known much more precisely than that.
➔ Please note the simulated ASY was nudged towards AERONET ASY data, so that we need to address the uncertainty in simulated ASY (not the uncertainty in the final ASY). Thus the suggested idea of perturbing final ASY by 0.05 is not a good idea. The old manuscript states "We do not address the model dependence on ASY or CAl_AE". If the reviewer meant perturbing the simulated ASY by ±0.05, we would like to say that such an experiment is too unrealistic. For example, AERONET data shows very similar ASY values over much of the ocean where sea salt aerosols dominate. The simulated ASY we generated also shows this feature. In other words, artificially increasing or decreasing by 0.05 will distort this feature. We wanted to address the uncertainty in simulated ASY within the extent to which the simulated ASY still looks reasonable.

14. p. 4. "Global simulations" is a somewhat misleading title for Section 2.3. No global simulations are described here, just the assumptions about aerosol optical properties.
➔ This title is meant to be a contrast with the titles in section 2.1 and section 2.2. In other words, global simulations mean no use of observation. We used the GOCART simulation and this is global simulation.

15. p. 5, 8th line of Sect. 2.4. Is "vertically summed aerosol extinction" the same as AOD, or something else?
➔ To convert the aerosol extinction coefficient to AOD, one has to integrate the aerosol extinction coefficient vertically (which factors in the size and unit of $\Delta z$). We simply added up the aerosol extinction coefficients, and so this is not an AOD.

16. p. 5–6. Section 3 should contain more information about the input data for the MACR model. Currently, it describes some details related to low clouds but does not explain how, in the first place, clouds were treated and where the cloud data came from. I realize that this information might be found in Choi and Chung (2014), but the main points should be repeated here to make this paper stand alone.
➔ Good idea. Done as suggested.

17. p. 8, first line: The criteria for selecting dust-dominated Aeronet sites are not intuitively clear. For example, why is it required that the absorption AOD exceeds 0.03? Dust is relatively weakly absorbing, with a single-scattering albedo of 0.96 at 550 nm used in this paper.
➔ We tried various criteria, and found the combination we have adopted for the paper. The end results worked good since Fig. 4A shows dust-dominated places. We required the AAOD criterion in order to remove a good mixture of sea salt and dust.

18. p. 9, last paragraph: It is concluded that the direct radiative forcing of anthropogenic aerosols is close to zero. Another important result, which would deserve some more discussion, is that the anthropogenic aerosols strongly increase atmospheric absorption (by 3.64 W m$^{-2}$, according to Table 1) and decrease the net solar radiation at the surface (by 3.74 W m$^{-2}$ ). This suggests that the primary climatic impact of anthropogenic aerosols is not that of cooling the planet but reducing precipitation (less solar energy available for evaporation, and less latent heating needed for balancing radiative cooling). Reduced precipitation due to absorbing aerosols has of course been discussed earlier, e.g. Ramanathan and Carmichael (2008) and Ming et al. (2010).
➔ The reviewer is right in the role of anthropogenic aerosols. Precipitation reduction has been discussed in many past studies, and we don't feel we need to address it in our study that focuses on the TOA forcing.

19. p. 10: Where was the review paper by Chung (2012) published?
➔ Yes. We added the DOI information.

20. p. 16: Fig. 4. The dot size scale (1, 2, 3) related to the number of AERONET climatological data is uninformative. Please be more explicit.
➔ We clarified this info in adding details in the caption of Fig. 4. The revised sentence states "The dot size is proportional to the number of AERONET data from decadal means (2001-2010) for each calendar."

Referee #2
In this paper, "Global direct aerosol radiative forcing, as constrained by comprehensive observations", the authors use a suite of observations from satellite and in-situ platforms to determine the optical properties of aerosol, from which the direct radiative effect can be determined when incorporated with a radiative transfer model and some information from aerosol models. The authors use the fine mode aerosol without sea salt and dust as a proxy for anthropogenic aerosol, and find that the radiative effect is more positive that previous estimates, stating that it is close to zero.

Main points
One key issue throughout the paper is the distinction between direct radiaitve forcing (DRF) and direct radiative effect (DRE). Generally, aerosol radiative forcing is used when talking about the change in radiative effect relative to pre-industrial levels. Here the analysis is concerned with the direct radiative effect, no comparison to the preindustrial aerosol radiative effect is made (other than in the final section). For clarity, this needs correcting throughout the paper to only use radiative forcing where the anthropogenic component is being considered, and care also taken to make sure comparisons are not being made with numbers from other literature that is concerned with the aerosol radiative forcing. Furthermore, if the authors intend to consider the anthropogenic aerosol radiative forcing then I think a more detailed consideration of the pre-industrial aerosol composition is necessary. Otherwise the result of this paper is really the fine mode DRE from non-dust and sea salt aerosols, not the anthropogenic DRF.
➔ We apologize for using confusing terms in the paper. To reduce the confusion, we changed the title of the paper and terms throughout the paper. For example, the revised paper consistently uses a term "direct fine-mode aerosol radiative effect", and avoids a term "forcing". We also re-wrote section 6 to accurately compare our observation estimate to the IPCC estimate. See "The global aerosol direct radiative forcing estimate in the 5[th] IPCC report is -0.35 Wm$^{-2}$ (Myhre et al., 2013), and this includes a dust forcing of -0.10 Wm$^{-2}$. Thus, the IPCC estimate is that anthropogenic carbonaceous, sulfate and nitrate aerosols pose a radiative forcing of -0.25 Wm$^{-2}$, while our observational estimate of total (anthropogenic + natural) carbonaceous, sulfate and nitrate aerosol forcing is -0.11 Wm$^{-2}$. The anthropogenic fraction (or pre-industrial fraction) of carbonaceous, sulfate and nitrate aerosols is uncertain. Black carbon, the only warming aerosol species in carbonaceous aerosol (black carbon + organic aerosol), sulfate and nitrate aerosol is known to be more anthropogenic than organic aerosols are (Bond et al., 2011). If the anthropogenic fraction of black carbon is similar to that of nitrate and sulfate aerosol, the aerosol direct radiative forcing becomes < -0.11 Wm$^{-2}$ in our observational estimation, which means that aerosol direct forcing is less negative than the consensus as expressed in the 5[th] IPCC report."
To clarify, the old manuscript didn't use the fine mode aerosol without sea salt and dust as a proxy for anthropogenic aerosol. In fact, in the old manuscript, we did not give numbers for our estimate of aerosol direct forcing since we did not make that estimate. We apologize for this confusion. The revised manuscript is clearer in this regard. The revised paper focuses more on fine-mode aerosol direct effect estimates, as the revised paper title shows that.

My second main point relates to the attribution of the fine mode aerosol to the BC, OC and inorganic species...
It seems strange that the fine-mode aerosol radiative effect quite abruptly ends at the west coast of Africa (Figure 3b). If this estimate includes natural aerosol, as stated, then I would expect a

contribution from fine dust aerosol over the mid-Atlantic. Does the radiative effect from fine dust really drop away that fast? Figure 5b suggests that there is a significant negative effect in the mid-Atlantic from dust and sea-salt, is this really balanced by a positive impact of other aerosol in this region?

➔ As Fig. 3a show, the amount of fine-mode aerosols does not end abruptly at the west coast of Africa. The direct effect does, since the surface albedo changes dramatically from very high values (over desert) to very low values (over the ocean). Absorbing aerosols over highly reflective surfaces have more positive (or less negative) forcing than those over non-reflective surfaces. To help the readers in this regard, we added the following phase in the caption of Fig. 1: "as estimated by a radiation model that includes observationally-derived surface albedo."

Related to the above, it is not clear from Figure 3 if this is surface or TOA, so perhaps I'm conflating two different metrics here.

➔ To our knowledge, aerosol forcing refers to the forcing at TOA unless specified otherwise. To be clearer in this regard, we added "TOA" in figure captions.

More importantly, the authors allude to the strange positive forcing from the 'anthropogenic' fine aerosol over the Sahara being the result of northward transport of biomass burning aerosol and bright desert surfaces. This is a testable hypothesis: there should be a relatively clear seasonal cycle in the impact, with most biomass burning emissions in the boreal winter time in West Africa. Is this the case?

➔ The reviewer raised a very good question. We looked into this issue during the revision and learned that the positive biomass burning forcing estimate over the Sahara does also exist beyond boreal winter. So, we revised the manuscript in the following:
We added the following para in section 5: "Scaling the simulated dust FMF to match AERONET FMF over dust-dominated sites may lead to overestimated or underestimated dust FMF outside of dust dominated regions. Plus, dust-dominated regions still have non-dust particles, and thus the simulated dust FMF might still underestimate or overestimate dust FMF even over dust dominated regions. This is why we conduct sensitivity runs." We added the following phases in this sentence in section 6: "The forcing is also conspicuously positive over the Sahara (Fig. 6A), partly because biomass burning aerosols in the Sahel are advected northwards in boreal winter (Haywood et al., 2008)".

Also, do the regions of highest radiative effect in Africa in Figure 6a align with the brightest regions of the surface reflectance assumed? The regions of high DRE do align with the locations of high AOD in Figure 1A that are generally considered the result of strong dust sources at those locations. There is potential for a similar problem in the Middle East and the Asian deserts too (it looks like there is non-dust fine mode AOD in the Teklemakan desert, which would be strange). This might indicate misattribution of dust aerosol as non-dust fine mode aerosol. The authors make efforts to consider the uncertainties involved in the estimate, which is appreciated; however, it is not clear how well the sensitivity studies would account for this likely mis-attribution. In this case, any error would cause a more positive radiative effect from the fine mode aerosol. This is a key point of the paper, so I think it is important to understand any potential bias.

➔ As explained earlier, we revised the manuscript to admit potential errors in the estimated fine-mode dust and sea salt AOD. We also think that misattribution can be dust aerosol as non-dust

fine mode aerosol and also be non-dust fine-mode aerosol as dust (depending on region and season). Thus, we added the following sentence: "As mentioned in section 5, our estimate of fine-mode sea salt and dust aerosols might be too large or too small over some areas. Possible over-estimation or under-estimation is likely reduced in global average, and so we focus on global averages as shown in Table 1." Realizing that the estimated fine-mode dust and sea salt direct effect is not as robust as the estimated total fine-mode direct effect, we made the revised paper focus on fine-mode direct effect estimates. The revised title emphasizes the focus on fine-mode direct effect estimates. In our opinion, the presented fine-mode direct radiative effect estimates are the most observational estimates to date, and can be very useful in validating aerosol simulation based estimates.

Smaller points
I think the language in the final sentence of the abstract is too strong. Consider replacing "near-zero" with "closer to zero than previous estimates".
➔ The reviewer is right. The revised sentence is "This net arises from total (natural + anthropogenic) carbonaceous, sulfate and nitrate aerosols, which suggests that global direct aerosol radiative forcing be less negative than $-0.35$ Wm$^{-2}$."

In the final paragraph of the first section I think it is important that the authors state when they use model information in this study. Based on that description it seems like models are not used, but this is not really true. An expanded discussion here would allow a better description of how this study builds upon the other studies cited, related to the minimization of reliance on models.
➔ We added the following sentence in section 1: "There is some use of aerosol simulation to fill up observation gaps in our study but the use is highly limited, and we address the uncertainty due to the use of simulation." We also expanded section 1 overall to better introduce our study.

The single sentence after the Section 2 heading either needs expanding or removing.
➔ We expanded.

Do you have some metric of the nudging that is applied to MODIS and MISR that can be stated in the paper? The change in the slope perhaps?
➔ The nudging formula were developed and tested during Chung et al. (2005). We added the following sentence in section 2.1: "See Chung et al. (2005) and Lee and Chung (2013) for the visual effects of the nudging."

With the 'nudging' of the GOCART model to the AERONET observations, in regions for which there are no AERONET observations is the result essentially just the model? Were any other functional forms than the 1/dˆ4 used? That suggests a very rapid relaxation back to the model values. Do you have any maps or numbers to indicate how strongly this nudging impacts the SSA and the ASY?
➔ Again, the nudging formula (for SSA and ASY) were developed and tested during Chung et al. (2005). In that study (Chung et al. 2005), we tested 1/d^2 as well as 1/d^4. After Chung et al. (2005) was published, we realized that the description of SSA/ASY nudging was not clearly explained. So, we decided to give the equations in the present study. We thus added the following sentence in section 2: "Please note that these above equations (Eq. 1 & 2) were also used in Chung et al. (2005) but a clear explanation was not given in that study."

The equations on pg4 are not very clear, consider using an equation editor and symbols instead of simply writing as text.
➔ We improved the clarity during the revision.

The high FMF from models at dust sites, relative to observations (Figure 4b), may be an issue with dust emissions favoring too small sizes (Kok et al., 2011). The authors should probably mention this, unless those models have been updated to reflect the latest emission distributions.
➔ Thanks for giving us the paper by Kok (2011). Kok pointed out that models overestimate the dust FMF and we found it too in Fig. 4B. The revised manuscript mentions the paper by Kok; see "Fig. 4B suggests that models tend to over-estimate dust FMF, at least over dust-dominated places, as pointed out by Kok (2011)." Discussing why the models overestimate the dust FMF, as Kok discusses, is beyond the scope of our study; if we need to discuss this, we also need to discuss other model discrepancies.

At the end of the first paragraph in section 6, the way the numbers are displayed is unnecessarily confusing. I think the final range can be quoted alone.
➔ We rephrased that sentence to reduce confusion.

Section 6: "At least > -0.28 Wm$^{-2}$" please reword, e.g. "more positive than -0.28 Wm$^{-2}$", "unlikely to be less than -0.28 Wm$^{-2}$"
➔ We modified section 6 greatly to be more precise. For example, see the 3$^{rd}$ para in section 6: "The global aerosol direct radiative forcing estimate in the 5$^{th}$ IPCC report is -0.35 Wm$^{-2}$ (Myhre et al., 2013), and this includes a dust forcing of -0.10 Wm$^{-2}$. Thus, the IPCC estimate is that anthropogenic carbonaceous, sulfate and nitrate aerosols pose a radiative forcing of -0.25 Wm$^{-2}$, while our observational estimate of total (anthropogenic + natural) carbonaceous, sulfate and nitrate aerosol forcing is -0.11 Wm$^{-2}$. The anthropogenic fraction (or pre-industrial fraction) of carbonaceous, sulfate and nitrate aerosols is uncertain. Black carbon, the only warming aerosol species in carbonaceous aerosol (black carbon + organic aerosol), sulfate and nitrate aerosol is known to be more anthropogenic than organic aerosols are (Bond et al., 2011). If the anthropogenic fraction of black carbon is similar to that of nitrate and sulfate aerosol, the aerosol direct radiative forcing becomes < -0.11 Wm$^{-2}$ in our observational estimation, which means that aerosol direct forcing is less negative than the consensus as expressed in the 5$^{th}$ IPCC report."

I'm not sure about the final sentence of the paper. There is still some reliance on models, albeit less than previous studies. Please consider revising the closing remarks.
➔ This request is quite difficult to accommodate. If we are very precise here (by mentioning a limited use of aerosol simulation), the overall point here will be diluted. We added the word "pure" so that the last sentence reads as "At least, our observational approach offers an independent estimate than pure aerosol simulations."

Table 1 - break the first column into two so that the property e.g. fine-mode DRE, can be displayed in the first column and then the specifics of that case put in the second column for ease of reading.
➔ Done as suggested.

Fig 1b and Fig 3b - state whether surface or TOA Fig 1b says DRE correctly, but then Fig 3b is called the aerosol forcing, please correct this throughout the paper Fig 6 - would it be useful to show the surface fine mode DRE here as well, for completeness?
➔ We modified figures and figure captions to improve the clarity.

Bellouin et al. (2008) reference has n/a errors for page numbers
➔ The page number has no meaning since it goes by DOI number for this paper.  We have thus deleted the page info.

---

## Referee Report (RR1)

The authors have addressed most of my comments on the original version of the manuscript. Nevertheless, I think that while the paper seems otherwise useful and scientifically sound, it is still somewhat let down by lack of clarity in writing, especially in Section 2. Most of my remaining suggestions (most importantly, comment 2) are concerned with this.

**Minor comments**

1. lines 15 and 23: To be absolutely clear, you should use "anthropogenic direct radiative forcing" here (to help those readers who have not accustomed to the practice that "forcing" always refers to the anthropogenic part).

2. While some clarifications have been added to Sect. 2, it still took me a lot of effort before I (hopefully!) finally understood it. To make this easier, I think it would help substantially if the reader knew beforehand, what the data will be used for. Therefore, I recommend to start Sect. 2 with a "preamble" paragraph where this is explained, something like this (I hope it approximates the truth sufficiently well):

*In sections 4 and 5 and Table 1, aerosol direct radiative effects (DRE) will be computed for three cases: (i) for the total aerosol, (ii) for the fine mode, and (iii) for fine mode sea-salt and dust. The total and fine-mode AOD are based on observations, as explained in Sect. 2.1. The other aerosol optical properties needed for the DRE calculation are derived as follows:*

- *The asymmetry parameter (ASY), the single-scattering albedo (SSA) and the SSA Ångström exponent (CAl_AE) for the total aerosol are derived by nudging GOCART simulated values towards AERONET data (Sect. 2.2). The spectral dependence of ASY is addressed as in Chung et al. (2005).*

- *The fine-mode aerosol DRE is computed as the difference between the total and coarse mode DREs. The coarse-mode ASY, SSA and CAl_AE are derived from GOCART simulations, as reported in Sect. 2.3.*

- *For computing the DRE due to fine-model sea-salt and dust, ASY, SSA and CAl_AE are derived from GOCART simulations (Sect. 2.3).*

*The datasets used to derive this information are explained in the following. All the datasets used in this study are monthly means.*

3. On lines 79–80, it is stated that coarse-mode AOD at 500 nm is obtained by subtracting fine-mode AOD from total AOD at 500 nm, and on lines 84–85 that the fine-mode AOD at 550 nm is obtained by subtracting the coarse-mode AOD at 500 nm from the [total] AOD at 550 nm. It is hard not to become puzzled here! Would it be clearer to say: *The fine-mode AOD at 550 nm is derived by adding the difference in total AOD between 550 and 500 nm to the fine-mode AOD at 500 nm, subject to the assumption that the coarse-mode AOD does not change between 500 and 550 nm?* Because this is what these sentences seem to imply:

$$fAOD_{550} = fAOD_{500} + (AOD_{550} - AOD_{500}) \tag{1}$$

4. On lines 89–90: "We apply a similar procedure to ASY (asymmetry parameter) and other aerosol optical properties". What does "other" include? Only CAl_AE?

5. Equations should be separated better from the text, (at least) on lines 114–117, 130, and 132.

6. Lines 144–147: if you accept my suggestion on the "preamble" of Sect. 2 (comment 2), this paragraph should be deleted. Indeed, currently it creates another potential confusion: "As for the optical properties (for example, SSA) for fine-mode aerosols, we do not need to address them ...", yet on lines 156–159, the optical properties for fine-mode dust and sea salt are discussed.

7. line 154: Delete "amount of" in "the amount of dust AOD".

8. On line 316: In math, $< -0.11$ means "more negative than $-0.11$". But is that what you want to say here? If you mean "less negative than $-0.11$", say exactly that.

9. line 334: Add paper number (D10205) for Bellouin et al. (2008).

10. Caption of Fig. 4. This should read "...for each calendar month". Furthermore, the dot size scale is equally uninformative as in the first version. Assuming that you have some quantitative limits to decide whether the dot size should be 1, 2 or 3, why not tell this to the reader?

11. Fig. 5, second panel: to be consistent with the terminology elsewhere, replace "Corresponding TOA forcing" with "Corresponding TOA radiative effect".

---

## Author Response (AR2)

-- Referee 1 --

The authors have addressed most of my comments on the original version of the manuscript. Nevertheless, I think that while the paper seems otherwise useful and scientifically sound, it is still somewhat let down by lack of clarity in writing, especially in Section 2. Most of my remaining suggestions (most importantly, comment 2) are concerned with this.
➔ We incorporated all the suggestions.

-- Referee 2 --

The authors have made efforts to improve the manuscript with extra text and shifting away from conclusions about the anthropogenic aerosol forcing; however, there are still some clarity issues. I see that the other reviewer also had trouble reading the paper. The additions unfortunately do not do much to improve this. In many places the language is opaque and additional statements like "overestimate or underestimate" and "higher or lower" do not allow concrete conclusions to be easily drawn. I've tried to cover some of the clarity issues below but this list is by no means exhaustive.

I'm still not fully convinced by the explanation for the large positive radiative effect observed over the Sahara (and Africa in general) after dust and sea-salt have been removed (Fig 6a). These are almost equal in magnitude to the DRE over the regions that are far more affected by biomass burning. It is also at odds with other estimates e.g. Bellouin et al. (2013; Fig. 4) and Heald et al. (2014; Fig 2). I realize that these are at least partially model-based, but the discrepancy in the TOA DRE is very large and needs to be addressed. The strong positive radiative effect at TOA shown in this paper suggests very little scattering impact of organic aerosol in the biomass burning aerosol. Other studies suggest the OA and BC contributions cancel to produce forcings closer to zero (e.g. Myhre et al., 2013) The results of this paper may be correct but they need contrasting with other studies and the reasons for differences explaining. It is worrying because the conclusion of the paper is that the radiative effect of the anthropogenic aerosol is less negative than previously thought. If there is a general overestimation of positive DRE then that might invalidate the conclusion. I don't think this is clearly explained and may not be adequately covered by the sensitivity tests.

➔ Fig. 4 of Bellouin et al. (2013) and Fig. 2 of Heald et al. (2014) show clear-sky aerosol radiative effect (which includes all the natural aerosols).  All of our aerosol radiative effect calculations include cloud.  (To be clear in this regard, we inserted "All the aerosol radiative effect estimates made by the MACR model in this study include 3D cloud effects" in Table 1 caption.)  Furthermore, Fig. 4 of Bellouin et al. (2013) and Fig. 2 of Heald et al. (2014) are about aerosol direct effect while Fig. 6a of our paper shows fine-mode aerosol radiative effect without

dust and sea salt. Thus, comparing Bellouin et al. (2013) and Heald et al. (2014) with our study is like comparing apple with orange. Having said this, the reviewer's concern about too positive TOA radiative effect in Fig. 6a of our study is understandable. However, please note that even Fig. 1B (total aerosol direct effect) shows positive forcing over the Saraha, while Fig. 4 of Bellouin et al. (2013) and Fig. 2 of Heald et al. (2014) show somewhat negative to neutral forcing. The differences between our study and their studies are multiple, one of which might be organic aerosol (OA) SSA. OA SSA is normally treated between 0.96 and 1.0 at 550nm but the true value is very uncertain due to brown carbon (e.g., Magi 2009 and 2011 estimated OA SSA to be 0.85). Our observation approach does not need to specify OA SSA while model based studies need to. OA SSA value can make very large impacts on the overall sign of biomass burning aerosol forcing. In view of this, we added and edited the $2^{nd}$ para of section 1. See "Plus, processing the calculated aerosol distribution by a radiation model requires the specification of parameters such as the single scattering albedo (SSA) of organic aerosol which has been treated as 0.96~1.0 at 550 nm in the modeling community (Myhre et al. 2013b) but might actually be much lower (e.g., 0.85 estimated by Magi 2009; 2011).". Again, we have given the most observational estimate of fine-mode aerosol forcing. Thus, the differences with model based or semi-empirical studies are obviously due to this methodology difference. Exploring the reasons for the result differences require another investigation leading to a separate paper.

The 5th assessment total aerosol radiative effect and forcing estimate is not entirely from models, as is somewhat implied in the final paragraphs of this paper. I'm sure the authors realize it does include partially-observational evaluation (e.g. Bellouin et al., 2013; Su et al., 2013) as well as the AEROCOM study that is used for individual components (Myhre et al. 2013), but this is not clear in the text. In the IPCC report, the authors do point out that both model and observations give similar results (see Chapter 7.5 and Chapter 8.3.4.2 Radiation Forcing of the Aerosol–Radiation Interaction by Component). The range on the direct aerosol effect given is -0.85 to +0.15 W/m2, encompassing the result of this study. Although this study has a narrower uncertainty I still find it hard to accept that this is a legitimate reduction rather than possibly not accounting for certain uncertainties (for example my point about the strong positive fine-mode

radiative effect that is surprising and potentially presents a positive bias in the direct aerosol effect).

➔ We are fully aware that the 5[th] IPCC report estimate includes semi-empirical studies.  We read our paper over and over again and cannot possibly see any implication that the 5[th] IPCC report estimate does not include these semi-empirical studies.  For example, we wrote "and these estimates depend heavily on aerosol simulation"; "The large spread among direct aerosol forcing estimates (Myhre et al., 2013a) is attributable largely to these simulation uncertainties".  These words ("heavily", "largely") imply that the IPCC estimates came mostly (but not entirely) from models.

Plus, in our final paragraph, we discussed the uncertainty of fine-mode forcing (instead of aerosol radiative forcing).  We did not claim that we reduced the uncertainty of aerosol direct radiative forcing estimates.

After reading it several times I feel that while the premise of the paper is interesting and the work has merit, I'm finding it difficult to recommend for publication in the current form because (1) I'm not fully convinced that the range in the uncertainty estimate encompasses the real uncertainty in this estimate, (2) the conclusions drawn about the less negative aerosol DRE are not robust as a result of the uncertainty, and (3) other readers are likely to struggle to understand the details of the paper considering both the reviewers here did. Furthermore, criticisms of the methodology in Lee and Chung (ACP, 2013), that is regularly cited in this work, were made by reviewers in the ACP discussions and do not seem to be fully addressed. I only bring this up because similar criticisms were raised during the initial review of this paper, so it may be worth the authors taking the time to better justify their methods and the uncertainties involved so that readers are convinced this methodology is robust.

➔ As we stated above, the reviewer did not clearly understand our paper.  As stated earlier, we have not discussed the uncertainty of aerosol direct forcing estimates in our first revised paper; instead we have discussed the uncertainty of fine-mode forcing estimates.  Regarding the methodology details, we improved the clarity during the revision.  The issues raised by the reviewers regarding Lee and Chung (2013) were either incorporated or rebutted but the editor did not send the rebuttal comments to the reviewers at the time.

I hope the authors understand that the criticisms here are made in order to improve the manuscript and to ensure that the conclusions are justified so that readers are convinced by the results.

Below are some minor comments.

pg2 ln 52 "simulated aerosol" or "modeled aerosol"

➔ Done as suggested.

pg2 ln52 the uncertainty isn't really "addressed", I would change that to "explored"

➔ The reviewer is mistaken.   We did address the uncertainty in used aerosol simulations. Again, we did not address the uncertainty of aerosol direct forcing estimates in our study, and that is not a focus of our revised study either (see the title).

pg2 ln79, multiple references to Lee and Chung (2013), here I think you should reference O'Neill et al. (2003) as that is the source of the AERONET SDA.

➔ We added the reference by O'Neill et al. (2003).

pg3 ln89, the Chin et al. reference is in a strange location, please alter.

➔ This was by design.  Chin et al. (2002) give the AOD for each aerosol species but doesn't give the SSA.

pg4 ln 104, I dont think the "but a clear explanation..." is necessary here.

➔ We inserted this phrase in response to the request by the other reviewer.

pg4 ln155-117, I still think that these would benefit to be written in an equation editor, with symbols for the AOD (either "AOD" ot tau) and subscript used for the species types.

➔ This is about the style, and a use of equation editor might create confusion (since regular texts and equation editors use different font styles).  However, we made each equation start each sentence to improve the clarity during the revision.

pg8 ln241, "SPRINTARS"

➔ There is no mistake here.

pg9 ln286-289, this is very confusing and does not really add any information unfortunately.

➔ We inserted this para in response to the other reviewer's concern but we understand possible confusion here.  To improve the clarity, we revised this para.  See "Scaling the simulated dust FMF to match AERONET FMF over dust-dominated sites may still have an overestimation or

underestimation of dust FMF outside of dust dominated regions.  Plus, dust-dominated regions have non-dust particles, and thus the scaled dust FMF might still underestimate or overestimate dust FMF even over dust dominated regions.  This is why we conduct sensitivity runs even after the scaling of the simulated dust FMF."   This para helps to understand the uncertainty of Fig. 6 and so we prefer to retain it.

pg10 ln309, the uncertainty range of -0.85 to +0.15 W/m2 should be quoted here for clarity

➔ Here, we meant to discuss the consensus.  We revised this sentence.  See "The consensus of global aerosol direct radiative forcing as shown in the 5$^{th}$ IPCC report"

pg10 ln316, "< -0.11 W/m2" means more negative, not closer to zero. I don't think this is what the authors actually mean.

➔ This mistake was also pointed by the other reviewer.  It is now corrected.

pg10 Figure 6 - year/period needs adding to the caption

➔ We added the period in the caption.